# Exploring the Promise and Limits of Real-Time Recurrent Learning

**Kazuki Irie**[1][†]  **Anand Gopalakrishnan**[2]  **Jürgen Schmidhuber**[2,3]
[1]Center for Brain Science, Harvard University, Cambridge, MA, USA
[2]The Swiss AI Lab, IDSIA, USI & SUPSI, Lugano, Switzerland
[3]AI Initiative, KAUST, Thuwal, Saudi Arabia
`kirie@fas.harvard.edu`,{`anand, juergen`}`@idsia.ch`

## Abstract

Real-time recurrent learning (RTRL) for sequence-processing recurrent neural networks (RNNs) offers certain conceptual advantages over backpropagation through time (BPTT). RTRL requires neither caching past activations nor truncating context, and enables online learning. However, RTRL's time and space complexity make it impractical. To overcome this problem, most recent work on RTRL focuses on approximation theories, while experiments are often limited to diagnostic settings. Here we explore the practical promise of RTRL in more realistic settings. We study actor-critic methods that combine RTRL and policy gradients, and test them in several subsets of DMLab-30, ProcGen, and Atari-2600 environments. On DMLab memory tasks, our system trained on fewer than $1.2\,\mathrm{B}$ environmental frames is competitive with or outperforms well-known IMPALA and R2D2 baselines trained on $10\,\mathrm{B}$ frames. To scale to such challenging tasks, we focus on certain well-known neural architectures with element-wise recurrence, allowing for tractable RTRL without approximation. Importantly, we also discuss rarely addressed limitations of RTRL in real-world applications, such as its complexity in the multi-layer case.[1]

## 1  Introduction

There are two classic learning algorithms to compute exact gradients for sequence-processing recurrent neural networks (RNNs): real-time recurrent learning (RTRL; Williams & Zipser (1989b;a); Robinson & Fallside (1987); Robinson (1989)) and backpropagation through time (BPTT; Rumelhart et al. (1986); Werbos (1990), reviewed in Sec. 2). In practice, BPTT is the only one commonly used today, simply because BPTT is tractable while RTRL is not. In fact, the time and space complexities of RTRL for a fully recurrent NN are quartic and cubic in the number of hidden units, respectively, which are prohibitive for any RNNs of practical sizes in real applications. Despite such an obvious complexity bottleneck, RTRL has certain attractive conceptual advantages over BPTT. BPTT requires to cache activations for each new element of the sequence processed by the model, for later gradient computation. As the amount of these past activations to be stored grows linearly with the sequence length, practitioners (constrained by the actual memory limit of their hardware) use the so-called *truncated* BPTT (TBPTT; Williams & Peng (1990)) where they specify the maximum number of time steps for this storage, giving up gradient components—and therefore credit assignments—that go beyond this time span. In contrast, RTRL does not require storing past activations, and enables computation of untruncated gradients for sequences of any arbitrary length. In addition, RTRL is an online learning algorithm (more efficient than BPTT to process long sequences in the online scenario) that allows for updating weights immediately after consuming every new input (assuming that the external error feedback to the model output is also available for each input). These attractive advantages of RTRL still actively motivate certain researchers to work towards practical RTRL (e.g., Tallec & Ollivier (2018); Mujika et al. (2018); Benzing et al. (2019); Menick et al. (2021); Silver et al. (2022)), while others are also motivated by the biological plausibility (e.g., Bellec et al. (2019); Lemmel & Grosu (2023)).

The root of RTRL's high complexities is the computation and storage of the so-called *sensitivity matrix* whose entries are derivatives of the hidden activations w.r.t. each trainable parameter of the

---

[1]Our code is public: `https://github.com/IDSIA/rtrl-elstm` [†]Work done at IDSIA.

model involved in the recurrence (see Sec. 2). Most recent research on RTRL focuses on introducing *approximation methods* into the computation and storage of this matrix. For example, Menick et al. (2021) introduce sparsity in both the weights of the RNN and updates of the temporal Jacobian (which is an intermediate matrix needed to compute the sensitivity matrix). Another line of work (Tallec & Ollivier, 2018; Mujika et al., 2018; Benzing et al., 2019) proposes estimators based on low-rank decompositions of the sensitivity matrix that are less expensive to compute and store than the original one. Silver et al. (2022) explore projection-based approximations of the sensitivity. The main research question in these lines of work is naturally focused around the quality of the proposed approximation method. Consequently, the central goal of their experiments is typically to test hyper-parameters and configurational choices that control the approximation quality in diagnostic settings, rather than evaluating the full potential of RTRL in realistic tasks. In the end, we still know very little about the true empirical promise of RTRL. Also, assuming that a solution is found to the complexity bottleneck, what actual applications or algorithms would RTRL unlock? In what scenarios would it be beneficial to replace BPTT by RTRL in today's deep learning?

Here we propose to study RTRL by looking ahead beyond research on approximations. We explore the full potential of RTRL in the settings where no approximation is needed, while at the same time, not restricting ourselves to toy tasks. For that, we focus on special RNN architectures with element-wise recurrence, that allow for tractable RTRL without any approximation. In fact, the quartic/cubic complexities of the fully recurrent NNs can be simplified for certain neural architectures. Many well-known RNN architectures, such as Quasi-RNNs (Bradbury et al., 2017) and Simple Recurrent Units (Lei et al., 2018), and even certain *linear Transformers* (Katharopoulos et al., 2020; Schmidhuber, 1991; Schlag et al., 2021), belong to this class of models (see Sec. 3.1). Note that the core idea underlying this observation is technically not new: Mozer (1989; 1991) and Gori et al. (1989) already explore an RNN architecture with this property in the late 1980s (even though the problematic multi-layer case is ignored; we discuss it in Sec. 5), and Javed et al. (2021; 2023) also exploit this in the architectural design of their RNNs. While such special RNNs may suffer from limited computational capabilities on certain tasks (i.e., one can come up with a synthetic/algorithmic task where such models fail; see Appendix B.2), they also often perform on par with fully recurrent NNs on many tasks (at least, this is the case for the tasks we explore in our experiments). For the purpose of this work, the RTRL-tractability property outweighs the potentially limited computational capabilities: these architectures allow us to focus on evaluating RTRL on challenging tasks with a scale that goes beyond the one typically used in prior RTRL work, and to draw conclusions without worrying about the quality of approximation. We study an actor-critic algorithm (Konda & Tsitsiklis, 1999; Sutton et al., 1999; Mnih et al., 2016) that combines RTRL and recurrent policy gradients (Wierstra et al., 2010), allowing credit assignments throughout an entire episode in reinforcement learning (RL) with partially observable Markov decision processes (POMDPs; Kaelbling et al. (1998); Schmidhuber (1990)). We test the resulting algorithm, Real-Time Recurrent Actor-Critic method (R2AC), in several subsets of DMLab-30 (Beattie et al., 2016), ProcGen (Cobbe et al., 2020), and Atari 2600 (Bellemare et al., 2013) environments, with a focus on memory tasks. In particular, on two memory environments of DMLab-30, our system is competitive with or outperforms the well-known IMPALA (Espeholt et al., 2018) and R2D2 (Kapturowski et al., 2019) baselines, demonstrating certain practical benefits of RTRL at scale. Finally, working with concrete real-world tasks also sheds lights on further limitations of RTRL that are rarely (if not never) discussed in prior work. These observations are important for future research on practical RTRL. We highlight and discuss these general challenges of RTRL (Sec. 5).

## 2 BACKGROUND

Here we first review real-time recurrent learning (RTRL; Williams & Zipser (1989b;a); Robinson & Fallside (1987); Robinson (1989)), which is a gradient-based learning algorithm for sequence-processing RNNs—an alternative to the now standard BPTT.

**Preliminaries.** Let $t$, $T$, $N$, and $D$ be positive integers. We describe the corresponding learning algorithm for the following standard RNN architecture (Elman, 1990) that transforms an input $\boldsymbol{x}(t) \in \mathbb{R}^D$ to an output $\boldsymbol{h}(t) \in \mathbb{R}^N$ at every time step $t$ as

$$\boldsymbol{s}(t) = \boldsymbol{W}\boldsymbol{x}(t) + \boldsymbol{R}\boldsymbol{h}(t-1) \quad ; \quad \boldsymbol{h}(t) = \sigma(\boldsymbol{s}(t)) \tag{1}$$

where $\boldsymbol{W} \in \mathbb{R}^{N \times D}$ and $\boldsymbol{R} \in \mathbb{R}^{N \times N}$ are trainable parameters, $\boldsymbol{s}(t) \in \mathbb{R}^N$, and $\sigma$ denotes the element-wise sigmoid function (we omit biases). For the derivation, it is convenient to describe each

component $s_k(t) \in \mathbb{R}$ of vector $s(t)$ for $k \in \{1, ..., N\}$,

$$s_k(t) = \sum_{n=1}^{D} W_{k,n} x_n(t) + \sum_{n=1}^{N} R_{k,n} \sigma(s_n(t-1)) \tag{2}$$

In addition, we consider some loss function $\mathcal{L}^{\text{total}}(1, T) = \sum_{t=1}^{T} \mathcal{L}(t) \in \mathbb{R}$ computed on an arbitrary sequence of length $T$ where $\mathcal{L}(t) \in \mathbb{R}$ is the loss at each time step $t$, which is a function of $h(t)$ (we omit writing explicit dependencies over the model parameters). Importantly, we assume that $\mathcal{L}(t)$ can be computed *solely* from $h(t)$ at step $t$ (i.e., $\mathcal{L}(t)$ has no dependency on any other past activations apart from $h(t-1)$ which is needed to compute $h(t)$).

The role of a gradient-based learning algorithm is to efficiently compute the gradients of the loss w.r.t. the trainable parameters of the model, i.e., $\dfrac{\partial \mathcal{L}^{\text{total}}(1, T)}{\partial W_{i,j}} \in \mathbb{R}$ for all $i \in \{1, ..., N\}$ and $j \in \{1, ..., D\}$, and $\dfrac{\partial \mathcal{L}^{\text{total}}(1, T)}{\partial R_{i,j}} \in \mathbb{R}$ for all $i, j \in \{1, ..., N\}$. RTRL and BPTT differ in the way to compute these quantities. While we focus on RTRL here, for the sake of completeness, we also provide an analogous derivation for BPTT in Appendix A.3.

**Real-Time Recurrent Learning (RTRL).** RTRL can be derived by first decomposing the total loss $\mathcal{L}^{\text{total}}(1, T)$ over time, and then summing all derivatives of each loss component $\mathcal{L}(t)$ w.r.t. intermediate variables $s_k(t)$ for all $k \in \{1, ..., N\}$:

$$\frac{\partial \mathcal{L}^{\text{total}}(1, T)}{\partial W_{i,j}} = \sum_{t=1}^{T} \frac{\partial \mathcal{L}(t)}{\partial W_{i,j}} = \sum_{t=1}^{T} \left( \sum_{k=1}^{N} \frac{\partial \mathcal{L}(t)}{\partial s_k(t)} \times \frac{\partial s_k(t)}{\partial W_{i,j}} \right) \tag{3}$$

In fact, unlike BPTT that can only compute the derivative of the total loss $\mathcal{L}^{\text{total}}(1, T)$ efficiently, RTRL is an online algorithm that computes each term $\dfrac{\partial \mathcal{L}(t)}{\partial W_{i,j}}$ through the decomposition above. The first factor $\dfrac{\partial \mathcal{L}(t)}{\partial s_k(t)}$ can be straightforwardly computed through standard backpropagation (as stated above, we assume there is no recurrent computation between $s(t)$ and $\mathcal{L}(t)$). For the second factor $\dfrac{\partial s_k(t)}{\partial W_{i,j}}$, which is an element of the so-called *sensitivity matrix/tensor*, we can derive a *forward recursion formula*, which can be obtained by directly differentiating Eq. 2:

$$\frac{\partial s_k(t)}{\partial W_{i,j}} = x_j(t) \mathbb{1}_{k=i} + \sum_{n=1}^{N} R_{k,n} \sigma'(s_n(t-1)) \frac{\partial s_n(t-1)}{\partial W_{i,j}} \tag{4}$$

where $\mathbb{1}_{k=i}$ denotes the indicator function: $\mathbb{1}_{k=i} = 1$ if $k = i$, and 0 otherwise, and $\sigma'$ denotes the derivative of sigmoid, i.e, $\sigma'(s_n(t-1)) = \sigma(s_n(t-1))(1 - \sigma(s_n(t-1)))$. The derivation is similar for $\dfrac{\partial \mathcal{L}(t)}{\partial R_{i,j}}$ where we obtain a recurrent formula to compute $\dfrac{\partial s_k(t)}{\partial R_{i,j}}$. As this algorithm requires to store $\dfrac{\partial s_k(t)}{\partial W_{i,j}}$ and $\dfrac{\partial s_k(t)}{\partial R_{i,j}}$, its space complexity is $O(N^2(D+N)) \sim O(N^3)$. The time complexity to update the sensitivity matrix/tensor via Eq. 4 is $O(N^4)$. To be fair with BPTT, it should be noted that $O(N^4)$ is the complexity for one update; this means that the time complexity to process a sequence of length $T$ is $O(TN^4)$.

Thanks to the forward recursion, the update frequency of RTRL is flexible: one can opt for *fully online learning*, where we update the weights using $\dfrac{\partial \mathcal{L}(t)}{\partial W}$ at every time step, or accumulate gradients for several time steps. It should be noted that frequent updates may result in *staleness* of the sensitivity matrix, as it accumulates updates computed using old weights (Eq. 4).

Note that algorithms similar to RTRL have been derived from several independent authors (see, e.g., Robinson & Fallside (1987); Kuhn et al. (1990); Gori et al. (1989); Mozer (1989), or Pearlmutter (1989); Gherrity (1989) for the continuous-time version).

## 3 METHOD

Our main algorithm is an actor-critic method that combines RTRL with recurrent policy gradients, using a special RNN architecture that allows for tractable RTRL. Here we describe its main components: an element-wise LSTM with tractable RTRL (Sec. 3.1), and the actor-critic algorithm that builds upon IMPALA (Espeholt et al., 2018) (Sec. 3.2).

### 3.1 RTRL FOR LSTM WITH ELEMENT-WISE RECURRENCE (eLSTM)

The main RNN architecture we use in this work is a variant of long short-term memory (LSTM; (Hochreiter & Schmidhuber, 1997)) RNN with *element-wise recurrence*. Let $\odot$ denote element-wise multiplication. At each time step $t$, it first transforms an input vector $\boldsymbol{x}(t) \in \mathbb{R}^D$ to a recurrent hidden state $\boldsymbol{c}(t) \in \mathbb{R}^N$ as follows:

$$\boldsymbol{f}(t) = \sigma(\boldsymbol{F}\boldsymbol{x}(t) + \boldsymbol{w}^f \odot \boldsymbol{c}(t-1)) \quad ; \quad \boldsymbol{z}(t) = \tanh(\boldsymbol{Z}\boldsymbol{x}(t) + \boldsymbol{w}^z \odot \boldsymbol{c}(t-1)) \tag{5}$$
$$\boldsymbol{c}(t) = \boldsymbol{f}(t) \odot \boldsymbol{c}(t-1) + (1 - \boldsymbol{f}(t)) \odot \boldsymbol{z}(t) \tag{6}$$

where $\boldsymbol{f}(t) \in \mathbb{R}^N$, $\boldsymbol{z}(t) \in \mathbb{R}^N$ are activations, $\boldsymbol{F} \in \mathbb{R}^{N \times D}$ and $\boldsymbol{Z} \in \mathbb{R}^{N \times D}$ are trainable weight matrices, and $\boldsymbol{w}^f \in \mathbb{R}^N$ and $\boldsymbol{w}^z \in \mathbb{R}^N$ are trainable weight vectors. These operations are followed by a gated feedforward NN to obtain an output $\boldsymbol{h}(t) \in \mathbb{R}^N$ as follows:

$$\boldsymbol{o}(t) = \sigma(\boldsymbol{O}\boldsymbol{x}(t) + \boldsymbol{W}^o \boldsymbol{c}(t)); \quad \boldsymbol{h}(t) = \boldsymbol{o}(t) \odot \boldsymbol{c}(t) \tag{7}$$

where $\boldsymbol{O} \in \mathbb{R}^{N \times D}$ and $\boldsymbol{W}^o \in \mathbb{R}^{N \times N}$ are trainable weight matrices. This architecture can be seen as an extension of Quasi-RNN (Bradbury et al., 2017) with element-wise recurrence in the gates, or Simple Recurrent Units (Lei et al., 2018) without depth gating, and also relates to many others (Li et al., 2018; Mozer, 1989; Gonnet & Deselaers, 2020; Gupta et al., 2022; Orvieto et al., 2023). While one could further discuss myriads of architectural details (Greff et al., 2016), most of them are irrelevant to our discussion on the complexity reduction in RTRL; the only essential property here is that "recurrence" is element-wise (whose limitations we discuss in Appendix B.2). We use this simple architecture above, an LSTM with element-wise recurrence (or eLSTM), for all our experiments.

Furthermore, we restrict ourselves to the one-layer case (we discuss the multi-layer case later in Sec. 5), where we assume that there is no recurrence after this layer. Based on this assumption, gradients for the parameters $\boldsymbol{O}$ and $\boldsymbol{W}^o$ in Eq. 7 can be computed by standard backpropagation, as they are not involved in recurrence. Hence, the sensitivity matrices we need for RTRL (Sec. 2) are: $\dfrac{\partial \boldsymbol{c}(t)}{\partial \boldsymbol{F}}, \dfrac{\partial \boldsymbol{c}(t)}{\partial \boldsymbol{Z}} \in \mathbb{R}^{N \times N \times D}$, and $\dfrac{\partial \boldsymbol{c}(t)}{\partial \boldsymbol{w}^f}, \dfrac{\partial \boldsymbol{c}(t)}{\partial \boldsymbol{w}^z} \in \mathbb{R}^{N \times N}$. Through trivial derivations (we provide the full derivation in Appendix A.1), we can show that each of these sensitivity matrices can be computed using a tractable forward recursion formula. For example for $\dfrac{\partial \boldsymbol{c}(t)}{\partial \boldsymbol{F}}$, we have, for $i, k \in \{1, ..., N\}$ and $j \in \{1, ..., D\}$,

$$\hat{\boldsymbol{f}}_i(t) = (\boldsymbol{c}_i(t-1) - \boldsymbol{z}_i(t))\boldsymbol{f}_i(t)(1 - \boldsymbol{f}_i(t)) \quad ; \quad \hat{\boldsymbol{z}}_i(t) = (1 - \boldsymbol{f}_i(t))(1 - \boldsymbol{z}_i(t)^2) \tag{8}$$
$$\hat{\boldsymbol{c}}_i(t) = \boldsymbol{f}_i(t) + \boldsymbol{w}_i^f \hat{\boldsymbol{f}}_i(t) + \boldsymbol{w}_i^z \hat{\boldsymbol{z}}_i(t) \tag{9}$$
$$\frac{\partial \boldsymbol{c}_i(t)}{\partial \boldsymbol{F}_{i,j}} = \hat{\boldsymbol{f}}_i(t)\boldsymbol{x}_j(t) + \hat{\boldsymbol{c}}_i(t)\frac{\partial \boldsymbol{c}_i(t-1)}{\partial \boldsymbol{F}_{i,j}} \; ; \text{ and } \quad \frac{\partial \boldsymbol{c}_k(t)}{\partial \boldsymbol{F}_{i,j}} = 0 \quad \text{for all } k \neq i. \tag{10}$$

where we introduce three intermediate vector $\hat{\boldsymbol{f}}(t), \hat{\boldsymbol{z}}(t), \hat{\boldsymbol{c}}(t) \in \mathbb{R}^N$ with components $\hat{\boldsymbol{f}}_i(t), \hat{\boldsymbol{z}}_i(t), \hat{\boldsymbol{c}}_i(t) \in \mathbb{R}$. Consequently, the gradients for the weights can be computed as:

$$\frac{\partial \mathcal{L}(t)}{\partial \boldsymbol{F}_{i,j}} = \sum_{k=1}^{N} \frac{\partial \mathcal{L}(t)}{\partial \boldsymbol{c}_k(t)} \times \frac{\partial \boldsymbol{c}_k(t)}{\partial \boldsymbol{F}_{i,j}} = \frac{\partial \mathcal{L}(t)}{\partial \boldsymbol{c}_i(t)} \times \frac{\partial \boldsymbol{c}_i(t)}{\partial \boldsymbol{F}_{i,j}} \tag{11}$$

**Finally**, we can compactly summarise these equations using standard matrix operations. By introducing notations $\hat{\boldsymbol{F}}(t) \in \mathbb{R}^{N \times D}$ with $\hat{\boldsymbol{F}}_{i,j}(t) = \dfrac{\partial \boldsymbol{c}_i(t)}{\partial \boldsymbol{F}_{i,j}} \in \mathbb{R}$, and $\boldsymbol{e}(t) \in \mathbb{R}^N$ with $\boldsymbol{e}_i(t) = \dfrac{\partial \mathcal{L}(t)}{\partial \boldsymbol{c}_i(t)} \in \mathbb{R}$

for $i \in \{1, ..., N\}$ and $j \in \{1, ..., D\}$, Eqs. 8-11 above can be written as:

$$\hat{\boldsymbol{f}}(t) = (\boldsymbol{c}(t-1) - \boldsymbol{z}(t)) \odot \boldsymbol{f}(t) \odot (1 - \boldsymbol{f}(t)) \quad ; \quad \hat{\boldsymbol{z}}(t) = (1 - \boldsymbol{f}(t)) \odot (1 - \boldsymbol{z}(t)^2) \quad (12)$$

$$\hat{\boldsymbol{c}}(t) = \boldsymbol{f}(t) + \boldsymbol{w}^f \odot \hat{\boldsymbol{f}}(t) + \boldsymbol{w}^z \odot \hat{\boldsymbol{z}}(t) \quad (13)$$

$$\hat{\boldsymbol{F}}(t) = \hat{\boldsymbol{f}}(t) \otimes \boldsymbol{x}(t) + \mathrm{diag}(\hat{\boldsymbol{c}}(t))\hat{\boldsymbol{F}}(t-1) \quad ; \quad \frac{\partial \mathcal{L}(t)}{\partial \boldsymbol{F}} = \mathrm{diag}(\boldsymbol{e}(t))\hat{\boldsymbol{F}}(t) \quad (14)$$

where, for notational convenience, we introduce a function $\mathrm{diag} : \mathbb{R}^N \to \mathbb{R}^{N \times N}$ that constructs a diagonal matrix whose diagonal elements are those of the input vector; however, in practical implementations (e.g., in PyTorch), this can be directly handled as vector-matrix multiplications with broadcasting (this is an important note for complexity analysis). $\otimes$ denotes outer-product. We can derive analogous equations for other parameters $\boldsymbol{Z}$, $\boldsymbol{w}^f$ and $\boldsymbol{w}^z$ (as well as biases which are omitted here) using the same intermediate variables ($\hat{\boldsymbol{f}}(t), \hat{\boldsymbol{z}}(t), \hat{\boldsymbol{c}}(t)$; Eqs. 12-13); see Appendix A.1.

The RTRL algorithm above requires maintaining sensitivity matrices $\hat{\boldsymbol{F}}(t) \in \mathbb{R}^{N \times D}$, and analogously defined $\hat{\boldsymbol{Z}}(t) \in \mathbb{R}^{N \times D}$, $\hat{\boldsymbol{w}}^f(t) \in \mathbb{R}^N$, and $\hat{\boldsymbol{w}}^z(t) \in \mathbb{R}^N$ (see Appendix A.1); thus, the space complexity is $O(DN) \sim O(N^2)$. The per-step time complexity is $O(N^2)$ (see Eqs. 8-11). Hence, this algorithm is tractable. Importantly, these equations 12-14 can be implemented as simple PyTorch code (just like the forward pass of the same model; Eqs. 5-7) without any non-standard logics. Note that many approximations of RTRL involve computations that are not well supported yet in standard deep learning libraries (e.g., efficiently handling custom sparsity in Menick et al. (2021)), which is an extra barrier for scaling RTRL through these methods in practice.

Note that the derivation of RTRL for element-wise recurrent nets is not novel: similar methods can be found in Mozer (1989; 1991); Gori et al. (1989) from the late 1980s. This result itself is also not very surprising, since element-wise recurrence introduces obvious sparsity in the temporal Jacobian (which is part of the second term in Eq. 4). Netherthless, we are not aware of any prior work pointing out that several modern RNN architectures such as Quasi-RNN (Bradbury et al., 2017) or Simple Recurrent Units (Lei et al., 2018) yield tractable RTRL (in the one-layer case). Also, while this is not the focus of our experiments, we show an example of linear Transformers/Fast Weight Programmers (Katharopoulos et al., 2020; Schmidhuber, 1991; Schlag et al., 2021) that have tractable RTRL (details can be found in Appendix A.2), which is another conceptually interesting result. We also note that the famous LSTM-algorithm (Hochreiter & Schmidhuber, 1997) (companion learning algorithm for the LSTM architecture) is a diagonal approximation of RTRL, so is the more recent SnAp-1 of Menick et al. (2021). Unlike in these works, the gradients computed by our RTRL algorithm above are *exact* for our eLSTM architecture. This allows us to draw conclusions from experimental results without worrying about the potential influence of approximation quality. We can evaluate the full potential of RTRL for this specific architecture.

Finally, this is also an interesting system from the biological standpoint. Each weight in the weight matrix/synaptic connections (e.g., $\boldsymbol{F} \in \mathbb{R}^{N \times D}$) is augmented with the corresponding "memory" ($\hat{\boldsymbol{F}}(t) \in \mathbb{R}^{N \times D}$) tied to its own learning process, which is updated in an online fashion, as the model observes more and more examples, through an Hebbian/outer product-based update rule (Eq. 14/Left).

## 3.2 Real-Time Recurrent Actor-Critic Policy Gradient Algorithm (R2AC)

The main algorithm we study in this work, Real-Time Recurrent Actor-Critic method (R2AC), combines RTRL with recurrent policy gradients. Our algorithm builds upon IMPALA (Espeholt et al., 2018). Essentially, we replace the RNN architecture and its learning algorithm, LSTM/TBPTT in the standard recurrent IMPALA algorithm, by our eLSTM/RTRL (Sec. 3.1). While we refer to the original paper (Espeholt et al., 2018) for basic details of IMPALA, here we recapitulate some crucial aspects. Let $M$ denote a positive integer. IMPALA is a distributed actor-critic algorithm where each *actor* interacts with the environment for a fixed number of steps $M$ to obtain a state-action-reward trajectory segment of length $M$ to be used by the *learner* to update the model parameters. $M$ is an important hyper-parameter that is used to specify the number of steps $M$ for $M$-step TD learning (Sutton & Barto, 1998) of the critic, and the frequency of weight updates. Given the same number of environmental steps used for training, systems trained with a smaller $M$ apply more weight updates than those trained with a higher $M$. For recurrent policies trained with TBPTT, $M$ also represents the BPTT span (i.e., BPTT is carried out on the $M$-length trajectory segment; no gradient is propagated

farther than $M$ steps back in time; while the last state of the previous segment is used as the initial state of the new segment in the forward pass). In the case of RTRL, there is no gradient truncation, but since $M$ controls the update frequency, the greater the $M$, the less frequently we update the parameters, and it potentially suffers less from sensitivity matrix staling. This setting allows for comparing TBPTT and RTRL in the setting where everything is equal (including the number of updates) except the actual gradients applied to the weights: truncated vs. untruncated ones.

Note that for R2AC with $M = 1$, one could obtain a *fully online* recurrent actor-critic method. However, in practice, it is known that $M > 1$ is crucial (for TD learning of the critic) for optimal performance. In all our experiments, we have $M > 1$. Our main focus is to evaluate learning with untruncated gradients, rather than the potential for online learning (discussed in Appendix B.3).

## 4    EXPERIMENTS

Here we present the main experiments of this work: RL in POMDPs using realistic game environments requiring memory. Other synthetic/diagnostic experiments are reported in Appendix B.1/B.2.

**DMLab Memory Tasks.**    DMLab-30 (Beattie et al., 2016) is a collection of 30 first-person 3D game environments, with a mix of both memory and reactive tasks. Here we focus on two well-known environments, `rooms_select_nonmatching_object` and `rooms_watermaze`, which are both categorised as "memory" tasks according to Parisotto et al. (2020). The mean episode lengths of these tasks are about 100 and 1000 steps, respectively. As we apply an action repetition of 4, each "step" corresponds to 4 environmental frames here. We refer to Appendix B.3 for further experimental details. Our model architecture is based on that of IMPALA (Espeholt et al., 2018). Both RTRL and TBPTT systems use our eLSTM (Sec. 3.1) as the recurrent layer with a hidden state size of 512. Everything is equal between these two systems except that the gradients are truncated in TBPTT but not in RTRL. To reduce the overall compute needed for the experiments, we first pre-train a TBPTT model for 50 M steps for `rooms_select_nonmatching_object`, and for 200 M steps for `rooms_watermaze`. Then, for all main training runs in this experiment, we initialise the parameters of the convolutional vision module from the same pre-trained model, and keep these parameters frozen (and thus, only train the recurrent layer and everything above it). For these main training runs, we train for 30 M and 100 M steps for `rooms_select_nonmatching_object` and `rooms_watermaze`, respectively; resulting in the total of 320 M and 1.2 B environmental frames. We compare RTRL and TBPTT for different values of $M \in \{10, 50, 100\}$ which influences: the weight update frequency, $M$-step TD learning, and for TBPTT, the backpropagation span (Sec. 3.2).

Table 1 shows the corresponding scores, and the left and middle parts of Figure 1 show the training curves. We observe that for `select_nonmatching_object` which has a short mean episode length of 100 steps, the performance of TBPTT and RTRL is similar even with $M = 50$. The benefit of RTRL is only visible in the case with $M = 10$. In contrast, for the more challenging `rooms_watermaze` task with the mean episode length of 1000 steps, RTRL outperforms TBPTT for all values of $M \in \{10, 50, 100\}$. Furthermore, with $M = 50$ or $100$, our RTRL system outperforms the IMPALA and R2D2 systems from prior work (Kapturowski et al., 2019), while trained on fewer than 1.2 B frames. Note that R2D2 systems (Kapturowski et al., 2019) are trained without action repetitions, and with a BPTT span of 80. This effectively demonstrates the practical benefit of RTRL in a realistic task requiring long-span credit assignments.

In Appendix B.3, we also show that RTRL does not have any unexpected negative impacts in a mostly reactive task (by testing our system on one environment of DMLab-30, `room_keys_doors_puzzle`, which is categorised as a reactive task according to Parisotto et al. (2020)). There, we also report practical computational costs, and discuss online learning.

**ProcGen.**    We test R2AC in another domain: ProcGen (Cobbe et al., 2020). We focus on one environment in the standard *hard-mode*, *Chaser*, which shows clear benefits of recurrent policies over those without memory (further details in Appendix B.4). *Chaser* is similar to the classic game "Pacman," effectively requiring some counting capabilities to fully exploit *power pellets* valid for a limited time span. The mean episode length for this task is about 200 steps, where each step is an environmental frame as we apply no action repeat for ProcGen. Unlike in the DMLab experiments above, here we train all models from scratch for 200 M steps without pre-training the vision module

Table 1: Scores on two memory environments of **DMLab-30**. Numbers on the top part are copied from the respective papers for reference. For both `rooms_select_nonmatching_object` and `rooms_watermaze`, we report mean and standard deviation computed over 3 training seeds (each using 3 sets of 100 test episodes; see Appendix B.3). "frames" indicates the number of environmental frames used for training. $M$ is the hyper-parameter that controls weight update frequency, $M$-step TD learning, and backpropagation span for TBPTT in IMPALA (see Sec. 3.2).

| | frames | $M$ | select_nonmatch | watermaze |
|---|---|---|---|---|
| IMPALA ((Espeholt et al., 2018)) | 1 B | 100 | 7.3 | 26.9 |
| IMPALA ((Kapturowski et al., 2019)) | 10 B | 100 | 39.0 | 47.0 |
| R2D2 ((Kapturowski et al., 2019)) | 10 B | - | 2.3 | 45.9 |
| R2D2+ ((Kapturowski et al., 2019)) | 10 B | - | 63.6 | 49.0 |
| TBPTT | < 1.2B | 10 | $54.5 \pm 1.1$ | $15.8 \pm 0.9$ |
| RTRL | | | $\mathbf{61.8} \pm 0.5$ | $\mathbf{40.2} \pm 5.6$ |
| TBPTT | < 1.2B | 50 | $61.4 \pm 0.5$ | $44.5 \pm 1.5$ |
| RTRL | | | $\mathbf{62.0} \pm 0.4$ | $\mathbf{52.3} \pm 1.9$ |
| TBPTT | < 1.2B | 100 | $61.7 \pm 0.1$ | $45.6 \pm 4.7$ |
| RTRL | | | $\mathbf{62.2} \pm 0.3$ | $\mathbf{54.8} \pm 4.3$ |

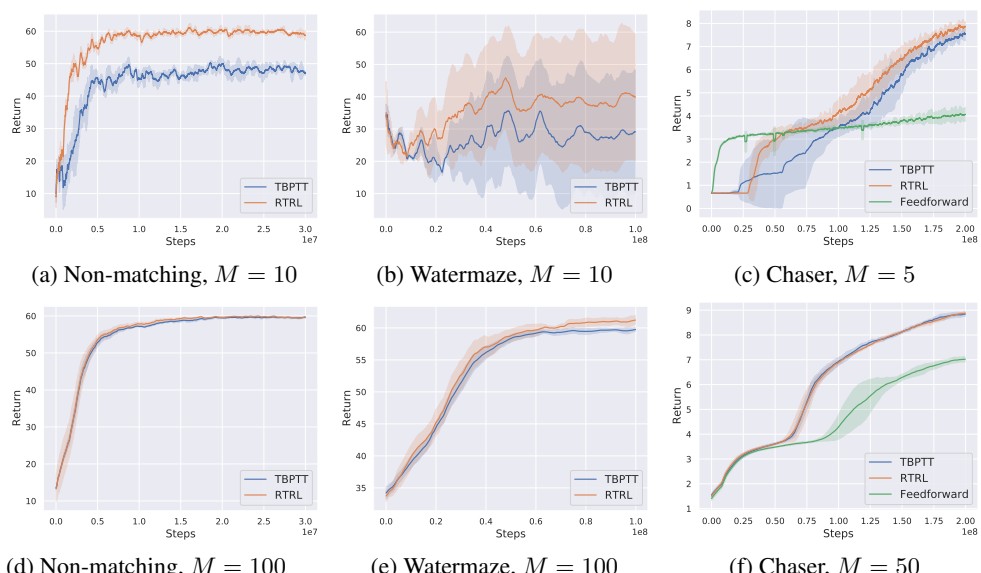

(a) Non-matching, $M = 10$     (b) Watermaze, $M = 10$     (c) Chaser, $M = 5$

(d) Non-matching, $M = 100$     (e) Watermaze, $M = 100$     (f) Chaser, $M = 50$

Figure 1: Training curves on **DMLab-30** `rooms_select_nonmatching_object` (Non-matching) and `rooms_watermaze` (Watermaze), and **Procgen** *Chaser* environments.

(since training the vision parameters using RTRL is intractable, they are trained with truncated gradients, i.e., only the recurrent layer is trained using RTRL; further discussed in Sec. 5). We compare RTRL and TBPTT with $M = 5$ or 50. The training curves are shown in the right part of Figure 1. Similar to the `rooms_select_nonmatching` case above, with a sufficiently large $M = 50$, there is no difference between RTRL and TBPTT, while we observe benefits of RTRL when $M = 5$.

**Atari.**     Apart from some exceptions (such as *Solaris* (Kapturowski et al., 2019)), many of the Atari game environments are considered to be fully observable when observations consist of a stack of 4 frames (Mnih et al., 2015; Hausknecht & Stone, 2015). However, it is also empirically known that, for certain games, recurrent policies yield higher performance than the feedforward ones having only access to 4 past frames (see, e.g., Mott et al. (2019); Kapturowski et al. (2019); Irie et al. (2021)). Here our general goal is to compare RTRL to TBPTT more broadly. We use five Atari environments: *Breakout*, *Gravitar*, *MsPacman*, *Q\*bert*, and *Seaquest*, following Kapturowski et al. (2019)'s selection

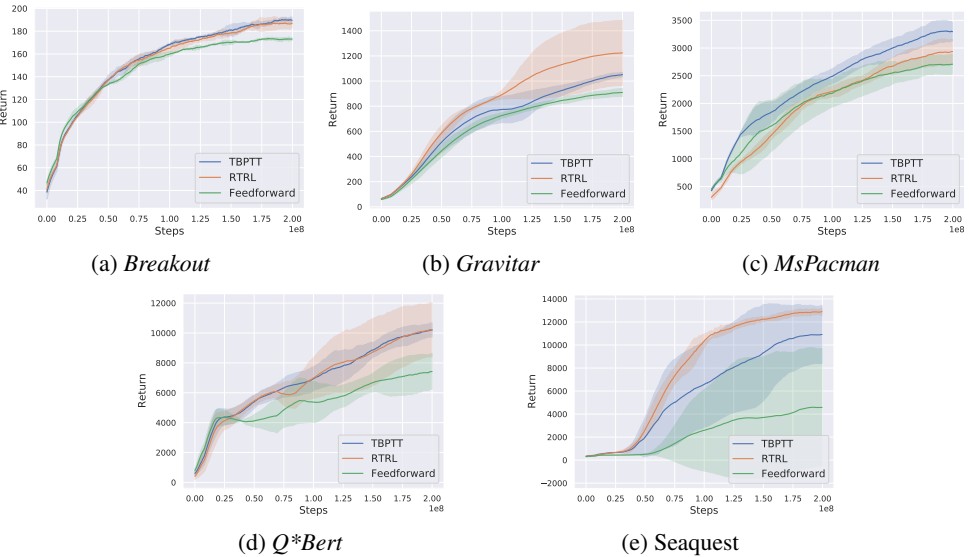

Figure 2: Learning curves on five **Atari** environments

for evaluating their R2D2. Here we use $M = 50$, and train for 200 M steps (with the action repeat of 4) from scratch. The learning curves are shown in Figure 2. With the exception of *MsPacman* (note that, unlike ProcGen/*Chaser* above, 4-frame stacking is used) where we observe a slight performance degradation, RTRL performs equally well or better than TBPTT in all other environments.

**Comparison to other baselines.** Table 2 report comparisons to other architectures and learning algorithms on the DMLab-30 memory tasks. The top-part of Table 2 is a pure architecture comparison using TBPTT for all models. We denote by 'feLSTM' an intermediate architecture between the standard LSTM and our eLSTM, obtained by replacing the element-wise recurrence in Eq. 5 of eLSTM by the full recurrence (i.e, weight vectors $w^f$ and $w^z$ in Eq. 5 are replaced by weight matrices; see Appendix A.5 for full details). eLSTM outperforms LSTM, while feLSTM performs similarly. RTRL for feLSTM is not tractable as it contains the full recurrence. The bottom-part of Table 2 shows that our exact RTRL using eLSTM clearly outperforms an approximate RTRL, SnAp-1, applied to feLSTM.

Table 2: Scores on two memory environments of **DMLab-30** for $M = 100$ (see also Table 3).

| Architecture | Recurrence | Learning Algorithm | select_nonmatch | watermaze |
|---|---|---|---|---|
| LSTM | full | TBPTT | $61.7 \pm 0.5$ | $37.6 \pm 5.3$ |
| feLSTM | full | TBPTT | $61.5 \pm 0.4$ | $43.2 \pm 4.7$ |
| eLSTM | element-wise | TBPTT | $61.7 \pm 0.1$ | $45.6 \pm 4.7$ |
| feLSTM | full | SnAp-1 | $53.6 \pm 0.8$ | $27.7 \pm 3.2$ |
| eLSTM | element-wise | RTRL | $\mathbf{62.2} \pm 0.3$ | $\mathbf{54.8} \pm 4.3$ |

## 5 LIMITATIONS AND DISCUSSION

Here we discuss limitations of this work, which also shed light on more general challenges of RTRL. Other limitations of our setting can be found in Appendix B.2.

**Multi-layer case of our RTRL.** The most crucial limitation of our tractable-RTRL algorithm for element-wise recurrent nets (Sec. 3.1) is its restriction to the one-layer case. By stacking two such layers, the corresponding RTRL algorithm becomes intractable as we end up with the same complexity bottleneck as in fully recurrent networks. This is simply because by composing two such element-wise recurrent layers, we obtain an NN which contains some "fully-connected" transformations over time as a whole. This can be easily seen in the following equations. By introducing extra superscripts to denote the layer number, in a stack of two element-wise LSTM layers of Eqs. 5-6 (we remove the

output gate), we can express the recurrent state $c^{(2)}(t)$ of the second layer at step $t$ as a function of the recurrent state $c^{(1)}(t-1)$ of the first layer from the previous step as follows:

$$c^{(2)}(t) = f^{(2)}(t) \odot c^{(2)}(t-1) + (1 - f^{(2)}(t)) \odot z^{(2)}(t) \tag{15}$$

$$f^{(2)}(t) = \sigma(F^{(2)} c^{(1)}(t) + w^{f(2)} \odot c^{(2)}(t-1)) \tag{16}$$

$$= \sigma(F^{(2)} \left( f^{(1)}(t) \odot c^{(1)}(t-1) + (1 - f^{(1)}(t)) \odot z^{(1)}(t) \right) + ...) \tag{17}$$

$$= \sigma(F^{(2)} f^{(1)}(t) \odot c^{(1)}(t-1) + F^{(2)}(1 - f^{(1)}(t)) \odot z^{(1)}(t) + ...) \tag{18}$$

By looking at the first term of Eq. 15 and that of Eq. 18, one can see that there is full recurrence between $c^{(2)}(t)$ and $c^{(1)}(t-1)$ via $F^{(2)}$, which brings back the quartic/cubic time and space complexity for the sensitivity of the recurrent state in the second layer w.r.t. parameters of the first layers. This limitation is not discussed in prior work (Mozer, 1989; 1991).

**Complexity of multi-layer RTRL in general.** Generally speaking, RTRL for the multi-layer case (a standard setting in modern deep learning) is rarely discussed (except Meert & Ludik (1997) and more recently Javed et al. (2023); Zucchet et al. (2023)). There are two important remarks to be made here.

First of all, even in an NN with a single RNN layer, if there is a layer with trainable parameters whose output is connected to the input of the RNN layer, a sensitivity matrix needs to be computed and stored for each of these parameters. A good illustration is the policy net used in all our RL experiments where our eLSTM layer takes the output of a deep (feedforward) convolutional net (the vision stem) as input. As training this vision stem using RTRL requires dealing with the corresponding sensitivity matrix, which is intractable, we train/pretrain the vision stem using TBPTT (Sec. 4). This is an important remark for RTRL research in general. For example, approximation methods proposed for the single-layer case may not scale to the multi-layer case, e.g., to exploit sparsity in the policy net above, it is not enough to assume weight sparsity in the RNN layer, but also in the vision stem.

Second, the multi-layer case introduces more complexity growth to RTRL than to BPTT. Let $L$ denote the number of layers. We seemlessly use BPTT with deep RNNs, as its time and space complexity is linear in $L$. This is not the case for RTRL. With RTRL, for each recurrent layer, we need to store sensitivities of all parameters of all preceding layers. This implies that, for an $L$-layer RNN, parameters in the first layer require $L$ sensitivity matrices, $L-1$ for the second layer, ..., etc., resulting in $L + (L-1) + ... + 2 + 1 = L(L+1)/2$ sensitivity matrices to be computed and stored. Given that multi-layer NNs are crucial today, this remains a big challenge for practical RTRL research.

**Principled vs. practical solution.** Another important aspect of RTRL research is that many realistic memory tasks have actual dependencies/credit assignment paths that are shorter than the maximum BPTT span we can afford in practice. In our experiments, with the exception of DMLab `rooms_watermaze` (Sec. 4), no task actually absolutely *requires* RTRL in practice; TBPTT with a large span suffices. Future improvements of the hardware may give a further advantage to TBPTT; the *practical* (simple) solution offered by TBPTT might be prioritised over the *principled* (complex) RTRL solution for dealing with long-span credit assignments. This is also somewhat reminiscent of the Transformer vs. RNN discussion regarding sequence processing with limited vs. unlimited context.

**Sequence-level parallelism.** While our study focuses on evaluation of untruncated gradients, another potential benefit of RTRL is online learning. For most standard self/supervised sequence-processing tasks such as language modelling, however, modern implementations are optimised to exploit access to the "full" sequence, and to leverage parallel computation across the time axis (at least for training; further discussed in Appendix B.2). While some hybrid RTRL-BPTT approaches (Williams (1989); Schmidhuber (1992); Williams & Zipser (1995); see Appendix A.4) may still be able to exploit such a parallelism, fast online learning remains open engineering challenge even with tractable RTRL.

## 6 CONCLUSION

We demonstrate the empirical promise of RTRL in realistic settings. By focusing on RNNs with element-wise recurrence, we obtain tractable RTRL without approximation. We evaluate our reinforcement learning RTRL-based actor-critic in several popular game environments. In one of the challenging DMLab-30 memory environments, our system outperforms the well-known IMPALA and R2D2 baselines which use many more environmental steps. We also highlight general important limitations and further challenges of RTRL rarely discussed in prior work.

## ACKNOWLEDGEMENTS

We thank Róbert Csordás for helpful suggestions for the figures, and help with installing DMLab. This research was partially funded by ERC Advanced grant no: 742870, project AlgoRNN, and by Swiss National Science Foundation grant no: 200021_192356, project NEUSYM. We are thankful for hardware donations from NVIDIA and IBM. The resources used for this work were partially provided by Swiss National Supercomputing Centre (CSCS) project s1145 and d123.

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

## A DERIVATIONS

### A.1 RTRL FOR LSTM WITH ELEMENT-WISE RECURRENCE

In Sec. 3.1, we only show RTRL equations (Eqs. 14) for one weight matrix of eLSTM ($\boldsymbol{F}$ in Eq. 5). Here we provide complete equations and their derivations for all other parameters.

First of all, the exact eLSTM architecture used in all our experiments use biases $\boldsymbol{b}^f, \boldsymbol{b}^z \in \mathbb{R}^N$, i.e., Eqs. 5 are replaced by:

$$\boldsymbol{f}(t) = \sigma(\boldsymbol{F}\boldsymbol{x}(t) + \boldsymbol{w}^f \odot \boldsymbol{c}(t-1) + \boldsymbol{b}^f) \quad ; \quad \boldsymbol{z}(t) = \tanh(\boldsymbol{Z}\boldsymbol{x}(t) + \boldsymbol{w}^z \odot \boldsymbol{c}(t-1) + \boldsymbol{b}^z) \quad (19)$$

Therefore, in addition to $\dfrac{\partial \boldsymbol{c}(t)}{\partial \boldsymbol{F}}, \dfrac{\partial \boldsymbol{c}(t)}{\partial \boldsymbol{Z}} \in \mathbb{R}^{N \times N \times D}$, and $\dfrac{\partial \boldsymbol{c}(t)}{\partial \boldsymbol{w}^f}, \dfrac{\partial \boldsymbol{c}(t)}{\partial \boldsymbol{w}^z} \in \mathbb{R}^{N \times N}$, we also have $\dfrac{\partial \boldsymbol{c}(t)}{\partial \boldsymbol{b}^f}, \dfrac{\partial \boldsymbol{c}(t)}{\partial \boldsymbol{b}^z} \in \mathbb{R}^{N \times N}$ as the sensitivity matrices to be computed/stored for RTRL. Note that adding these biases do not change the RTRL equations for $\boldsymbol{F}$ presented in Eq. 5/Sec. 3.1.

We recall that we define $\boldsymbol{e}(t) \in \mathbb{R}^N$ with $\boldsymbol{e}_i(t) = \dfrac{\partial \mathcal{L}(t)}{\partial \boldsymbol{c}_i(t)} \in \mathbb{R}$ for $i \in \{1, ..., N\}$ in Sec. 3.1, which can be computed using standard backpropagation (as we assume that we have no recurrent layer between $\boldsymbol{c}(t)$ and $\mathcal{L}(t)$).

We also recall that, for convenience, we introduce three following intermediate variables $\hat{\boldsymbol{f}}(t), \hat{\boldsymbol{z}}(t), \hat{\boldsymbol{c}}(t) \in \mathbb{R}^N$ which appear in several equations.

$$\hat{\boldsymbol{f}}(t) = (\boldsymbol{c}(t-1) - \boldsymbol{z}(t)) \odot \boldsymbol{f}(t) \odot (1 - \boldsymbol{f}(t)) \quad ; \quad \hat{\boldsymbol{z}}(t) = (1 - \boldsymbol{f}(t)) \odot (1 - \boldsymbol{z}(t)^2) \quad (12)$$

$$\hat{\boldsymbol{c}}(t) = \boldsymbol{f}(t) + \boldsymbol{w}^f \odot \hat{\boldsymbol{f}}(t) + \boldsymbol{w}^z \odot \hat{\boldsymbol{z}}(t) \quad (13)$$

The complete list of our RTRL equations is as follows.

**Equations for $\boldsymbol{F}$.** We define $\hat{\boldsymbol{F}}(t) \in \mathbb{R}^{N \times D}$ with $\hat{\boldsymbol{F}}_{i,j}(t) = \dfrac{\partial \boldsymbol{c}_i(t)}{\partial \boldsymbol{F}_{i,j}} \in \mathbb{R}$.

$$\hat{\boldsymbol{F}}(t) = \hat{\boldsymbol{f}}(t) \otimes \boldsymbol{x}(t) + \operatorname{diag}(\hat{\boldsymbol{c}}(t))\hat{\boldsymbol{F}}(t-1) \quad ; \quad \dfrac{\partial \mathcal{L}(t)}{\partial \boldsymbol{F}} = \operatorname{diag}(\boldsymbol{e}(t))\hat{\boldsymbol{F}}(t) \quad (14)$$

**Equations for $\boldsymbol{Z}$.** We define $\hat{\boldsymbol{Z}}(t) \in \mathbb{R}^{N \times D}$ with $\hat{\boldsymbol{Z}}_{i,j}(t) = \dfrac{\partial \boldsymbol{c}_i(t)}{\partial \boldsymbol{Z}_{i,j}} \in \mathbb{R}$.

$$\hat{\boldsymbol{Z}}(t) = \hat{\boldsymbol{z}}(t) \otimes \boldsymbol{x}(t) + \operatorname{diag}(\hat{\boldsymbol{c}}(t))\hat{\boldsymbol{Z}}(t-1) \quad ; \quad \dfrac{\partial \mathcal{L}(t)}{\partial \boldsymbol{Z}} = \operatorname{diag}(\boldsymbol{e}(t))\hat{\boldsymbol{Z}}(t) \quad (20)$$

**Equations for $\boldsymbol{w}^f$.** We define $\hat{\boldsymbol{w}}^f(t) \in \mathbb{R}^N$ with $\hat{\boldsymbol{w}}^f_i(t) = \dfrac{\partial \boldsymbol{c}_i(t)}{\partial \boldsymbol{w}^f_i} \in \mathbb{R}$.

$$\hat{\boldsymbol{w}}^f(t) = \hat{\boldsymbol{f}}(t) \odot \boldsymbol{c}(t-1) + \hat{\boldsymbol{c}}(t) \odot \hat{\boldsymbol{w}}^f(t-1) \quad ; \quad \dfrac{\partial \mathcal{L}(t)}{\partial \boldsymbol{w}^f} = \boldsymbol{e}(t) \odot \hat{\boldsymbol{w}}^f(t) \quad (21)$$

**Equations for $\boldsymbol{w}^z$.** We define $\hat{\boldsymbol{w}}^z(t) \in \mathbb{R}^N$ with $\hat{\boldsymbol{w}}^z_i(t) = \dfrac{\partial \boldsymbol{c}_i(t)}{\partial \boldsymbol{w}^z_i} \in \mathbb{R}$.

$$\hat{\boldsymbol{w}}^z(t) = \hat{\boldsymbol{z}}(t) \odot \boldsymbol{c}(t-1) + \hat{\boldsymbol{c}}(t) \odot \hat{\boldsymbol{w}}^z(t-1) \quad ; \quad \dfrac{\partial \mathcal{L}(t)}{\partial \boldsymbol{w}^z} = \boldsymbol{e}(t) \odot \hat{\boldsymbol{w}}^z(t) \quad (22)$$

**Equations for $\boldsymbol{b}^f$.** We define $\hat{\boldsymbol{b}}^f(t) \in \mathbb{R}^N$ with $\hat{\boldsymbol{b}}^f_i(t) = \dfrac{\partial \boldsymbol{c}_i(t)}{\partial \boldsymbol{b}^f_i} \in \mathbb{R}$.

$$\hat{\boldsymbol{b}}^f(t) = \hat{\boldsymbol{f}}(t) + \hat{\boldsymbol{c}}(t) \odot \hat{\boldsymbol{b}}^f(t-1) \quad ; \quad \dfrac{\partial \mathcal{L}(t)}{\partial \boldsymbol{b}^f} = \boldsymbol{e}(t) \odot \hat{\boldsymbol{b}}^f(t) \quad (23)$$

**Equations for $b^z$.** We define $\hat{\boldsymbol{b}}^z(t) \in \mathbb{R}^N$ with $\hat{b}_i^z(t) = \dfrac{\partial c_i(t)}{\partial b_i^z} \in \mathbb{R}$.

$$\hat{\boldsymbol{b}}^z(t) = \hat{\boldsymbol{z}}(t) + \hat{\boldsymbol{c}}(t) \odot \hat{\boldsymbol{b}}^z(t-1) \quad ; \quad \frac{\partial \mathcal{L}(t)}{\partial \boldsymbol{b}^z} = \boldsymbol{e}(t) \odot \hat{\boldsymbol{b}}^z(t) \tag{24}$$

Note that the actual implementation is tested by comparing the gradients computed through our RTRL algorithm against those computed by BPTT (using the common PyTorch reverse-mode automatic differentiation).

Now we provide the corresponding derivations. Let $i, k \in \{1, ..., N\}$ and $j \in \{1, ..., D\}$.

**Derivation for $\boldsymbol{F}$.** Common to all cases, the goal is to compute the gradients of the loss w.r.t. the corresponding model parameter. Since we assume there is no recurrent layer after this layer (see our settings of Sec. 2), we can express the corresponding gradient as:

$$\frac{\partial \mathcal{L}(t)}{\partial \boldsymbol{F}_{i,j}} = \sum_{k=1}^{N} \frac{\partial \mathcal{L}(t)}{\partial \boldsymbol{c}_k(t)} \times \frac{\partial \boldsymbol{c}_k(t)}{\partial \boldsymbol{F}_{i,j}} \tag{25}$$

As we assume we can compute the first factor $\dfrac{\partial \mathcal{L}(t)}{\partial \boldsymbol{c}_k(t)} = \boldsymbol{e}_k(t)$ using standard backpropagation, what remains to be computed is the second factor $\dfrac{\partial \boldsymbol{c}_k(t)}{\partial \boldsymbol{F}_{i,j}}$. The direct differentiation of Eq. 6 yields:

$$\frac{\partial \boldsymbol{c}_k(t)}{\partial \boldsymbol{F}_{i,j}} = (\boldsymbol{c}_k(t-1) - \boldsymbol{z}_k(t)) \frac{\partial \boldsymbol{f}_k(t)}{\partial \boldsymbol{F}_{i,j}} + \boldsymbol{f}_k(t) \frac{\partial \boldsymbol{c}_k(t-1)}{\partial \boldsymbol{F}_{i,j}} + (1 - \boldsymbol{f}_k(t)) \frac{\partial \boldsymbol{z}_k(t)}{\partial \boldsymbol{F}_{i,j}} \tag{26}$$

Now, we explicitly compute $\dfrac{\partial \boldsymbol{f}_k(t)}{\partial \boldsymbol{F}_{i,j}}$ and $\dfrac{\partial \boldsymbol{z}_k(t)}{\partial \boldsymbol{F}_{i,j}}$ as:

$$\frac{\partial \boldsymbol{f}_k(t)}{\partial \boldsymbol{F}_{i,j}} = \boldsymbol{f}_k'(t) \left( \boldsymbol{x}_j(t) \mathbb{1}_{k=i} + \boldsymbol{w}_k^f \frac{\partial \boldsymbol{c}_k(t-1)}{\partial \boldsymbol{F}_{i,j}} \right) \; ; \; \frac{\partial \boldsymbol{z}_k(t)}{\partial \boldsymbol{F}_{i,j}} = \boldsymbol{z}_k'(t) \boldsymbol{w}_k^z \frac{\partial \boldsymbol{c}_k(t-1)}{\partial \boldsymbol{F}_{i,j}} \tag{27}$$

where $\boldsymbol{f}'(t), \boldsymbol{z}'(t) \in \mathbb{R}^N$ are the "derivatives" of $\boldsymbol{f}(t)$ (sigmoid) and $\boldsymbol{z}(t)$ (tanh), i.e.,

$$\boldsymbol{f}'(t) = \boldsymbol{f}(t) \odot (1 - \boldsymbol{f}(t)) \quad ; \quad \boldsymbol{z}'(t) = 1 - \boldsymbol{z}(t)^2 \tag{28}$$

Now by substituting Eqs. 27 in Eq. 26, we obtain the following forward recursion equation:

$$\frac{\partial \boldsymbol{c}_k(t)}{\partial \boldsymbol{F}_{i,j}} = (\boldsymbol{c}_k(t-1) - \boldsymbol{z}_k(t)) \boldsymbol{f}_k'(t) \boldsymbol{x}_j(t) \mathbb{1}_{k=i} \tag{29}$$

$$+ \left( \boldsymbol{f}_k(t) + (\boldsymbol{c}_k(t-1) - \boldsymbol{z}_k(t)) \boldsymbol{f}_k'(t) \boldsymbol{w}_k^f + (1 - \boldsymbol{f}_k(t)) \boldsymbol{z}_k'(t) \boldsymbol{w}_k^z \right) \frac{\partial \boldsymbol{c}_k(t-1)}{\partial \boldsymbol{F}_{i,j}} \tag{30}$$

Since the sensitivities are initialised by zero, i.e., $\dfrac{\partial \boldsymbol{c}_k(0)}{\partial \boldsymbol{F}_{i,j}} = 0$, and the additive term in Eq. 29 is non zero if and only if $k = i$, it follows that, for any $t$,

$$\frac{\partial \boldsymbol{c}_k(t)}{\partial \boldsymbol{F}_{i,j}} = 0 \quad \text{if} \quad k \neq i. \tag{31}$$

The other entries (the case where $k = i$) can be compactly represented using intermediate variables introduced above ($\hat{\boldsymbol{f}}(t), \hat{\boldsymbol{z}}(t), \hat{\boldsymbol{c}}(t)$ ; Eqs. 12-13), as follows:

$$\frac{\partial \boldsymbol{c}_i(t)}{\partial \boldsymbol{F}_{i,j}} = (\boldsymbol{c}_i(t-1) - \boldsymbol{z}_i(t)) \boldsymbol{f}_i'(t) \boldsymbol{x}_j(t)$$

$$+ \left( \boldsymbol{f}_i(t) + (\boldsymbol{c}_i(t-1) - \boldsymbol{z}_i(t)) \boldsymbol{f}_i'(t) \boldsymbol{w}_i^f + (1 - \boldsymbol{f}_i(t)) \boldsymbol{z}_i'(t) \boldsymbol{w}_i^z \right) \frac{\partial \boldsymbol{c}_i(t-1)}{\partial \boldsymbol{F}_{i,j}} \tag{32}$$

$$= \hat{\boldsymbol{f}}_i(t) \boldsymbol{x}_j(t) + \left( \boldsymbol{f}_i(t) + \hat{\boldsymbol{f}}_i(t) \boldsymbol{w}_i^f + \hat{\boldsymbol{z}}_i(t) \boldsymbol{w}_i^z \right) \frac{\partial \boldsymbol{c}_i(t-1)}{\partial \boldsymbol{F}_{i,j}} \tag{33}$$

$$= \hat{\boldsymbol{f}}_i(t) \boldsymbol{x}_j(t) + \hat{\boldsymbol{c}}_i(t) \frac{\partial \boldsymbol{c}_i(t-1)}{\partial \boldsymbol{F}_{i,j}} \tag{34}$$

Finally, by introducing the notation $\hat{\boldsymbol{F}}(t) \in \mathbb{R}^{N \times D}$ with $\hat{\boldsymbol{F}}_{i,j}(t) = \dfrac{\partial \boldsymbol{c}_i(t)}{\partial \boldsymbol{F}_{i,j}} \in \mathbb{R}$, we arrive at Eq. 14/Left. Eq. 14/Right for the loss gradients is obtained by simplifying Eq. 25 using Eq. 31 (the sum reduces to one term).

The derivation is similar for $\boldsymbol{Z}$. We now show the derivation for $\boldsymbol{w}^f$.

**Derivation for $\boldsymbol{w}^f$.** Starting over, the goal is to compute the following gradient:

$$\frac{\partial \mathcal{L}(t)}{\partial \boldsymbol{w}_i^f} = \sum_{k=1}^{N} \frac{\partial \mathcal{L}(t)}{\partial \boldsymbol{c}_k(t)} \times \frac{\partial \boldsymbol{c}_k(t)}{\partial \boldsymbol{w}_i^f} = \sum_{k=1}^{N} \boldsymbol{e}_k(t) \frac{\partial \boldsymbol{c}_k(t)}{\partial \boldsymbol{w}_i^f} \tag{35}$$

The direct differentiation through Eq. 6 yields:

$$\frac{\partial \boldsymbol{c}_k(t)}{\partial \boldsymbol{w}_i^f} = (\boldsymbol{c}_k(t-1) - \boldsymbol{z}_k(t)) \frac{\partial \boldsymbol{f}_k(t)}{\partial \boldsymbol{w}_i^f} + \boldsymbol{f}_k(t) \frac{\partial \boldsymbol{c}_k(t-1)}{\partial \boldsymbol{w}_i^f} + (1 - \boldsymbol{f}_k(t)) \frac{\partial \boldsymbol{z}_k(t)}{\partial \boldsymbol{w}_i^f} \tag{36}$$

Now, we explicitly compute $\dfrac{\partial \boldsymbol{f}_k(t)}{\partial \boldsymbol{w}_i^f}$ and $\dfrac{\partial \boldsymbol{z}_k(t)}{\partial \boldsymbol{w}_i^f}$, which yields:

$$\frac{\partial \boldsymbol{f}_k(t)}{\partial \boldsymbol{w}_i^f} = \boldsymbol{f}_k'(t) \left( \boldsymbol{c}_k(t-1) \mathbb{1}_{k=i} + \boldsymbol{w}_k^f \frac{\partial \boldsymbol{c}_k(t-1)}{\partial \boldsymbol{w}_i^f} \right) \;;\; \frac{\partial \boldsymbol{z}_k(t)}{\partial \boldsymbol{w}_i^f} = \boldsymbol{z}_k'(t) \boldsymbol{w}_k^z \frac{\partial \boldsymbol{c}_k(t-1)}{\partial \boldsymbol{w}_i^f} \tag{37}$$

Now by substituting Eqs. 37 in Eq. 36, we obtain the following forward recursion equation:

$$\frac{\partial \boldsymbol{c}_k(t)}{\partial \boldsymbol{w}_i^f} = (\boldsymbol{c}_k(t-1) - \boldsymbol{z}_k(t)) \boldsymbol{f}_k'(t) \boldsymbol{c}_k(t-1) \mathbb{1}_{k=i} \tag{38}$$

$$+ \left( \boldsymbol{f}_k(t) + (\boldsymbol{c}_k(t-1) - \boldsymbol{z}_k(t)) \boldsymbol{f}_k'(t) \boldsymbol{w}_k^f + (1 - \boldsymbol{f}_k(t)) \boldsymbol{z}_k'(t) \boldsymbol{w}_k^z \right) \frac{\partial \boldsymbol{c}_k(t-1)}{\partial \boldsymbol{w}_i^f} \tag{39}$$

Similar to the derivation for $\boldsymbol{F}$, as the sensitivities are initially zero, i.e., $\dfrac{\partial \boldsymbol{c}_k(t-1)}{\partial \boldsymbol{w}_i^f} = 0$, and non-zero terms are added only to the entries where $k = i$, it follows that for any $t$, $\dfrac{\partial \boldsymbol{c}_k(t)}{\partial \boldsymbol{w}_i^f} = 0$ if $k \neq i$, and the non-zero entries are:

$$\frac{\partial \boldsymbol{c}_i(t)}{\partial \boldsymbol{w}_i^f} = \hat{\boldsymbol{f}}_i(t) \boldsymbol{c}_i(t-1) + \hat{\boldsymbol{c}}_i(t) \frac{\partial \boldsymbol{c}_i(t-1)}{\partial \boldsymbol{w}_i^f} \tag{40}$$

which finally yields Eq. 21.

The derivation is almost the same for $\boldsymbol{b}^f$, except that, instead of $\boldsymbol{c}_k(t-1) \mathbb{1}_{k=i}$ in Eq. 37, we obtain $\mathbb{1}_{k=i}$. Finally, the derivations are also similar for $\boldsymbol{w}^z$ and $\boldsymbol{b}^z$.

## A.2 TRACTABLE RTRL FOR CERTAIN LINEAR TRANSFORMERS/FAST WEIGHT PROGRAMMERS

While the main focus of this work is to evaluate tractable RTRL using eLSTM, we also discuss that there are several neural architectures with tractable RTRL in the one-layer case. Here we show that RTRL is also tractable for a certain "simple" one-layer linear Transformers/Fast Weight Programmers (FWPs; Katharopoulos et al. (2020); Schmidhuber (1991); Schlag et al. (2021); Irie et al. (2021)).

The FWP in question transforms an input $\boldsymbol{x}(t) \in \mathbb{R}^N$ to an output $\boldsymbol{y}(t) \in \mathbb{R}^N$ at each time step $t$, as follows:

$$[\boldsymbol{k}(t), \boldsymbol{v}(t), \boldsymbol{q}(t)] = [\boldsymbol{K}\boldsymbol{x}(t), \boldsymbol{V}\boldsymbol{x}(t), \boldsymbol{Q}\boldsymbol{x}(t)] \tag{41}$$

$$\boldsymbol{W}(t) = \boldsymbol{W}(t-1) + \boldsymbol{v}(t) \otimes \boldsymbol{k}(t) \tag{42}$$

$$\boldsymbol{y}(t) = \boldsymbol{W}(t) \sigma(\boldsymbol{q}(t)) \tag{43}$$

where $\boldsymbol{k}(t) \in \mathbb{R}^N$, $\boldsymbol{v}(t) \in \mathbb{R}^N$, $\boldsymbol{q}(t) \in \mathbb{R}^N$, $\boldsymbol{W}(t) \in \mathbb{R}^{N \times N}$, with trainable parameters $\boldsymbol{K} \in \mathbb{R}^{N \times N}$, $\boldsymbol{V} \in \mathbb{R}^{N \times N}$, and $\boldsymbol{Q} \in \mathbb{R}^{N \times N}$ (we use the same dimension $N$ everywhere for simplicity, but the

following derivation remains valid for the general case where $\boldsymbol{x}(t)$ is of dimension $D$; the only requirement is that $\boldsymbol{k}(t)$ and $\boldsymbol{q}(t)$ are of the same dimension).

Let $i, j \in \{1, .., N\}$. Using the same definition of the loss $\mathcal{L}(t)$ defined in Sec. 2, the goal is to compute $\dfrac{\partial \mathcal{L}(t)}{\partial \boldsymbol{Q}_{i,j}}, \dfrac{\partial \mathcal{L}(t)}{\partial \boldsymbol{K}_{i,j}}, \dfrac{\partial \mathcal{L}(t)}{\partial \boldsymbol{V}_{i,j}}$. Since $\boldsymbol{q}(t)$ is not involved in the recurrent loop (here we are again under the one-layer assumption), the gradients $\dfrac{\partial \mathcal{L}(t)}{\partial \boldsymbol{Q}_{i,j}}$ can be computed using standard backpropagation. Hence, we focus on $\dfrac{\partial \mathcal{L}(t)}{\partial \boldsymbol{K}_{i,j}}$ and $\dfrac{\partial \mathcal{L}(t)}{\partial \boldsymbol{V}_{i,j}}$. Here we provide the RTRL derivation for $\dfrac{\partial \mathcal{L}(t)}{\partial \boldsymbol{K}_{i,j}}$ (the derivation for $\dfrac{\partial \mathcal{L}(t)}{\partial \boldsymbol{V}_{i,j}}$ is analogous). Let $l, m, i$ and $j$ denote positive integers. As in Sec. 2, it is practical to write down each element $\boldsymbol{W}_{l,m}(t) \in \mathbb{R}$ of $\boldsymbol{W}(t) \in \mathbb{R}^{N \times N}$, as well as each element $\boldsymbol{k}_m(t) \in \mathbb{R}$ of $\boldsymbol{k}(t) \in \mathbb{R}^N$, for all $l, m \in \{1, ..., N\}$:

$$\boldsymbol{W}_{l,m}(t) = \boldsymbol{W}_{l,m}(t-1) + \boldsymbol{v}_l(t)\boldsymbol{k}_m(t) \tag{44}$$

$$\boldsymbol{k}_m(t) = \sum_{n=1}^{N} \boldsymbol{K}_{m,n}\boldsymbol{x}_n(t) \tag{45}$$

The goal is to compute the following gradient:

$$\frac{\partial \mathcal{L}(t)}{\partial \boldsymbol{K}_{i,j}} = \sum_{l=1}^{N} \sum_{m=1}^{N} \frac{\partial \mathcal{L}(t)}{\partial \boldsymbol{W}_{l,m}(t)} \times \frac{\partial \boldsymbol{W}_{l,m}(t)}{\partial \boldsymbol{K}_{i,j}} \tag{46}$$

The first factor $\dfrac{\partial \mathcal{L}(t)}{\partial \boldsymbol{W}_{l,m}(t)}$ can be computed using standard backpropagation (no recurrence involved). We derive a forward recursion formula for the second factor by differentiating Eq. 44:

$$\frac{\partial \boldsymbol{W}_{l,m}(t)}{\partial \boldsymbol{K}_{i,j}} = \frac{\partial \boldsymbol{W}_{l,m}(t-1)}{\partial \boldsymbol{K}_{i,j}} + \boldsymbol{v}_l(t)\frac{\partial \boldsymbol{k}_m(t)}{\partial \boldsymbol{K}_{i,j}} \tag{47}$$

Now, differentiating Eq. 45 yields:

$$\frac{\partial \boldsymbol{k}_m(t)}{\partial \boldsymbol{K}_{i,j}} = \begin{cases} 0 & \text{if} \quad m \neq i \\ \dfrac{\partial \boldsymbol{k}_i(t)}{\partial \boldsymbol{K}_{i,j}} = \boldsymbol{x}_j(t) & \text{Otherwise} \end{cases} \tag{48}$$

This last equation is a source of complexity reduction: in the 3-dimentional tensor $\dfrac{\partial \boldsymbol{k}_m(t)}{\partial \boldsymbol{K}_{i,j}}$, many terms are zero, and non-zero component only depends on $j$. Eq. 47 becomes:

$$\frac{\partial \boldsymbol{W}_{l,m}(t)}{\partial \boldsymbol{K}_{i,j}} = \begin{cases} 0 & \text{if} \quad m \neq i \\ \dfrac{\partial \boldsymbol{W}_{l,i}(t-1)}{\partial \boldsymbol{K}_{i,j}} + \boldsymbol{v}_l(t)\boldsymbol{x}_j(t) & \text{Otherwise} \end{cases} \tag{49}$$

Since the term added at each step, $\boldsymbol{v}_l(t)\boldsymbol{x}_j(t)$, only depends on $l$ and $j$, by introducing a matrix $\hat{\boldsymbol{K}}(t) \in \mathbb{R}^{N \times N}$ such that, for all $l, j \in \{1, ..., N\}$, $\hat{\boldsymbol{K}}_{l,j}(t) = \dfrac{\partial \boldsymbol{W}_{l,i}(t)}{\partial \boldsymbol{K}_{i,j}}$ for all $i \in \{1, ..., N\}$, we can compactly express the 4-dimensional sensitivity tensor with elements $\dfrac{\partial \boldsymbol{W}_{l,m}(t)}{\partial \boldsymbol{K}_{i,j}}$ using the 2-dimensional sensitivity matrix with elements $\hat{\boldsymbol{K}}_{l,j}(t)$ through sparsity and parameter-sharing obtained as directly consequences of the forward computation of the FWP defined above. Hence, the space complexity of RTRL is reduced to $O(N^2)$. Now Eq. 46 also greatly simplifies:

$$\frac{\partial \mathcal{L}(t)}{\partial \boldsymbol{K}_{i,j}} = \sum_{l=1}^{N} \sum_{m=1}^{N} \frac{\partial \mathcal{L}(t)}{\partial \boldsymbol{W}_{l,m}(t)} \times \frac{\partial \boldsymbol{W}_{l,m}(t)}{\partial \boldsymbol{K}_{i,j}} = \sum_{l=1}^{N} \frac{\partial \mathcal{L}(t)}{\partial \boldsymbol{W}_{l,i}(t)} \times \frac{\partial \boldsymbol{W}_{l,i}(t)}{\partial \boldsymbol{K}_{i,j}} \tag{50}$$

By defining a matrix $\boldsymbol{E}(t) \in \mathbb{R}^{N \times N}$ such that $\boldsymbol{E}_{l,i}(t) = \dfrac{\partial \mathcal{L}(t)}{\partial \boldsymbol{W}_{l,i}(t)}$, Eq. 50 can be computed through one simple matrix multiplication; and Eq. 49 yields a simple forward recursion formula for $\hat{\boldsymbol{K}}(t)$:

$$\frac{\partial \mathcal{L}(t)}{\partial \boldsymbol{K}} = \boldsymbol{E}(t)^{\mathsf{T}} \hat{\boldsymbol{K}}(t) \quad ; \quad \hat{\boldsymbol{K}}(t) = \hat{\boldsymbol{K}}(t-1) + \boldsymbol{v}(t) \otimes \boldsymbol{x}(t) \tag{51}$$

where $\mathsf{T}$ denotes transpose. The time complexity of Eq. 51/Left (for one update) is $O(N^3)$ which is tractable. The batch version of Eq. 51/Left can be implemented as a *batch matrix-matrix multiplication* in standard deep learning libraries.

The derivation is analogous for $\boldsymbol{V}$ (by analogously defining $\hat{\boldsymbol{V}}(t)$) which yields:

$$\frac{\partial \mathcal{L}(t)}{\partial \boldsymbol{V}} = \boldsymbol{E}(t) \hat{\boldsymbol{V}}(t) \quad ; \quad \hat{\boldsymbol{V}}(t) = \hat{\boldsymbol{V}}(t-1) + \boldsymbol{k}(t) \otimes \boldsymbol{x}(t) \tag{52}$$

In summary, RTRL for this simple FWP is tractable with the time/space complexities of $O(N^3)$ and $O(N^2)$. As for eLSTM (Sec. 5), this result is only valid for the one-layer case. Also note that the standard FWP (see e.g., Irie et al. (2021)) applies softmax on both key and query vectors; here we need to remove such an activation function applied to the key (see Eq. 42) to make RTRL tractable.

**A "biological" memory system view.** The resulting system (consisting of the neural architecture, plus the RTRL learning algorithm) forms an interesting type of memory "organism." The system maintains two kinds of "synaptic" memories in an online fashion: the fast weight matrix $\boldsymbol{W}(t)$ is a short-term memory where the system stores information required to *solve the task* at hand (memory based on input observations; Eq. 42), while sensitivity matrices $\hat{\boldsymbol{K}}(t)$ and $\hat{\boldsymbol{V}}(t)$ (paired to the model's weight matrices $\boldsymbol{K}(t)$ and $\boldsymbol{V}(t)$) store another memory required for *learning* (memory based on external feedback to model outputs; Eqs. 51 and 52/Right).

**Remarks.** Strictly speaking, RTRL is the name of the learning algorithm when forward-mode automatic differentiation (AD) is applied to RNNs. Here we refer to RTRL as a generic name referring to the forward AD applied to any sequence models including linear Transformers. Regarding the standard Transformer: we note that a Transformer anyway needs to store all past activations even for its forward pass, and we are not aware of any straightforward RTRL-like online learning algorithm that leads to complexity reduction of any form (compared to real-time BPTT (Williams & Zipser, 1995)).

## A.3 BACKPROPAGATION THROUGH TIME (BPTT)

In Sec. 2, we review RTRL. For the sake of completeness, here we review BPTT using the same notations and settings described in Sec. 2.

BPTT can be obtained by directly summing derivatives of the *total loss* $\mathcal{L}^{\text{total}}(1, T)$ w.r.t. all intermediate variables $\boldsymbol{s}_k(t)$ for all $k \in \{1, ..., N\}$ and $t \in \{1, ..., T\}$:

$$\frac{\partial \mathcal{L}^{\text{total}}(1, T)}{\partial \boldsymbol{W}_{i,j}} = \sum_{t=1}^{T} \sum_{k=1}^{N} \frac{\partial \mathcal{L}^{\text{total}}(1, T)}{\partial \boldsymbol{s}_k(t)} \times \frac{\partial \boldsymbol{s}_k(t)}{\partial \boldsymbol{W}_{i,j}(t)} \tag{53}$$

where we introduce the notation $\dfrac{\partial \boldsymbol{s}_k(t)}{\partial \boldsymbol{W}_{i,j}(t)}$ denoting the derivative of $\boldsymbol{s}_k(t)$ w.r.t. the *variable* representing the weight $\boldsymbol{W}_{i,j}$ at time $t$, $\boldsymbol{W}_{i,j}(t)$ (meaning that $\boldsymbol{s}_k(t-1)$ is a constant w.r.t $\boldsymbol{W}_{i,j}(t)$ and thus, $\dfrac{\partial \boldsymbol{s}_k(t)}{\partial \boldsymbol{W}_{i,j}(t)}$ is only the first term of $\dfrac{\partial \boldsymbol{s}_k(t)}{\partial \boldsymbol{W}_{i,j}}$ in Eq. 4, i.e., $\dfrac{\partial \boldsymbol{s}_k(t)}{\partial \boldsymbol{W}_{i,j}(t)} = \boldsymbol{x}_j(t) \mathbb{1}_{k=i}$). This allows us to write gradients w.r.t. the weights at a specific time step, e.g., $\dfrac{\partial \mathcal{L}^{\text{total}}(1, T)}{\partial \boldsymbol{W}_{i,j}} = \sum_{\tau=1}^{T} \dfrac{\partial \mathcal{L}^{\text{total}}(1, T)}{\partial \boldsymbol{W}_{i,j}(\tau)}$.

By introducing the classic notation $\boldsymbol{\delta}(t) \in \mathbb{R}^N$ with $\boldsymbol{\delta}_k(t) = \dfrac{\partial \mathcal{L}^{\text{total}}(1, T)}{\partial \boldsymbol{s}_k(t)} \in \mathbb{R}$ for all $k \in \{1, ..., N\}$,

$$\boldsymbol{\delta}_k(t) = \sum_{\tau=1}^{T} \frac{\partial \mathcal{L}(\tau)}{\partial \boldsymbol{s}_k(t)} = \sum_{\tau=t}^{T} \frac{\partial \mathcal{L}(\tau)}{\partial \boldsymbol{s}_k(t)} = \frac{\partial \mathcal{L}(t)}{\partial \boldsymbol{s}_k(t)} + \sum_{\tau=t+1}^{T} \frac{\partial \mathcal{L}(\tau)}{\partial \boldsymbol{s}_k(t)} \tag{54}$$

$$= \frac{\partial \mathcal{L}(t)}{\partial \boldsymbol{s}_k(t)} + \sum_{n=1}^{N} \sum_{\tau=t+1}^{T} \frac{\partial \mathcal{L}(\tau)}{\partial \boldsymbol{s}_n(t+1)} \times \frac{\partial \boldsymbol{s}_n(t+1)}{\partial \boldsymbol{s}_k(t)} \tag{55}$$

Since $\displaystyle\sum_{\tau=t+1}^{T} \frac{\partial \mathcal{L}(\tau)}{\partial \boldsymbol{s}_n(t+1)} = \boldsymbol{\delta}_n(t+1)$ in Eq. 55, we obtain the backward recursion formula for $\boldsymbol{\delta}(t)$:

$$\boldsymbol{\delta}_k(t) = \frac{\partial \mathcal{L}(t)}{\partial \boldsymbol{s}_k(t)} + \sum_{n=1}^{N} \boldsymbol{\delta}_n(t+1) \times \frac{\partial \boldsymbol{s}_n(t+1)}{\partial \boldsymbol{s}_k(t)} = \frac{\partial \mathcal{L}(t)}{\partial \boldsymbol{s}_k(t)} + \sum_{n=1}^{N} \boldsymbol{\delta}_n(t+1) \boldsymbol{R}_{n,k} \sigma'(\boldsymbol{s}_k(t)) \tag{56}$$

$$\frac{\partial \mathcal{L}^{\text{total}}(1, T)}{\partial \boldsymbol{W}_{i,j}} = \sum_{t=1}^{T} \sum_{k=1}^{N} \boldsymbol{\delta}_k(t) \times \frac{\partial \boldsymbol{s}_k(t)}{\partial \boldsymbol{W}_{i,j}(t)} = \sum_{t=1}^{T} \boldsymbol{\delta}_i(t) \boldsymbol{x}_j(t) \tag{57}$$

In the end, all quantities involved in these equations can be expressed as matrix-matrix/vector multiplications, element-wise vector multiplications $\odot$, or outer products between two vectors $\otimes$:

$$\boldsymbol{\delta}(t) = \frac{\partial \mathcal{L}(t)}{\partial \boldsymbol{s}(t)} + (\boldsymbol{R}^{\mathsf{T}} \boldsymbol{\delta}(t+1)) \odot \sigma'(\boldsymbol{s}(t)) \quad ; \quad \frac{\partial \mathcal{L}^{\text{total}}(1, T)}{\partial \boldsymbol{W}} = \sum_{t=1}^{T} \boldsymbol{\delta}(t) \otimes \boldsymbol{x}(t) \tag{58}$$

The formula for $\dfrac{\partial \mathcal{L}^{\text{total}}(1, T)}{\partial \boldsymbol{R}}$ can be derived analogously. This is the derivation which is obtained as a "natural" extension of the standard backpropagation algorithm for feedforward networks, by unfolding the RNN over time. BPTT requires to store intermediate activations $\boldsymbol{s}(t) \in \mathbb{R}^N$ (and inputs $\boldsymbol{x}(t) \in \mathbb{R}^D$) for all $t \in \{1, .., T\}$, resulting in the space complexity of $O(T(N + D)) \sim O(TN)$. We can only compute the gradients $\dfrac{\partial \mathcal{L}^{\text{total}}(1, T)}{\partial \boldsymbol{W}}$ after going through the entire sequence (because of the backward recursion to compute $\boldsymbol{\delta}(t)$; Eq. 58/Left). The (per-update) time complexity is $O(TN^2)$ (thus, the corresponding per-step time complexity is $O(N^2)$).

Technically speaking, one could also adopt BPTT for online learning (by applying BPTT at each time step; *real-time BPTT* (Williams & Zipser, 1995)). However, this would result in many redundant computations, with a time complexity of order of $O(T^2 N^2)$ which is intractable in practice, and more expensive than RTRL's $O(N^4)$ for long sequences with $T > N$.

## A.4 A Hybrid BPTT-RTRL Method

As we discuss in Sec. 5, fully online learning is computationally sub-optimal for certain sequence processing applications as it prevents parallel computation over time. In fact, the sequence-level parallelism is an extra advantage of BPTT over RTRL that is not captured by the complexity analysis; even for fully recurrent NNs, many operations can be parallelised over the time dimension (e.g., the linear transformation of the input $\boldsymbol{x}(t)$ in Eq 1). When fully online learning is not crucial, a hybrid approach (Williams, 1989; Schmidhuber, 1992; Williams & Zipser, 1995) that combines BPTT and RTRL is more attractive than RTRL to compute untruncated gradients efficiently.

Such a hybrid method is a *segment-wise* online algorithm. The main idea is to chunk sequences into segments, and to process sequences by applying BPTT to each segment while carrying over sensitivities needed by RTRL at segment boundaries to compute untruncated gradients. Here we review Schmidhuber (1992)'s derivation in the setting of Sec. 2. Let $t_0$, $t_1$, $K$ and $S$ be positive integers. In addition to the notations already introduced above (including the notation $\boldsymbol{W}_{i,j}(t)$ that denotes the variable corresponding to weights $\boldsymbol{W}_{i,j}$ at specific time step $t$; introduced in Appendix A.3), we use $\mathcal{L}(t_0, t_1)$ to denote the sum of losses $\mathcal{L}(t)$ for $t$ from step $t_0$ to $t_1$ (both inclusive).

The algorithm iterates over segments of length $S$ that constitute the full sequence of length $T$ (naturally, the length of the last segment is shorter than or equal to $S$). At iteration step $K > 0$,

we assume that $K$ segments are already *processed*, i.e., by denoting $t_0 = K * S$, three quantities $\dfrac{\partial \mathcal{L}(1, t_0)}{\partial \boldsymbol{W}_{i,j}}$, $\boldsymbol{s}_i(t_0)$ and $\mathbf{W}_{k,i,j}(t_0) = \sum_{\tau=1}^{t_0} \dfrac{\partial \boldsymbol{s}_k(t_0)}{\partial \boldsymbol{W}_{i,j}(\tau)}$ are computed and cached for all $i, k \in \{1, ..., N\}$ and $j \in \{1, ..., D\}$. We process the next segment as follows:

$$\frac{\partial \mathcal{L}(1, t_0 + S)}{\partial \boldsymbol{W}_{i,j}} = \frac{\partial \mathcal{L}(1, t_0)}{\partial \boldsymbol{W}_{i,j}} + \frac{\partial \mathcal{L}(t_0 + 1, t_0 + S)}{\partial \boldsymbol{W}_{i,j}} \tag{59}$$

The first term is computed by the previous iteration. We need to compute the second term:

$$\frac{\partial \mathcal{L}(t_0 + 1, t_0 + S)}{\partial \boldsymbol{W}_{i,j}} = \sum_{\tau=1}^{t_0+S} \frac{\partial \mathcal{L}(t_0 + 1, t_0 + S)}{\partial \boldsymbol{W}_{i,j}(\tau)} \tag{60}$$

$$= \sum_{\tau=1}^{t_0} \frac{\partial \mathcal{L}(t_0 + 1, t_0 + S)}{\partial \boldsymbol{W}_{i,j}(\tau)} + \sum_{\tau=t_0+1}^{t_0+S} \frac{\partial \mathcal{L}(t_0 + 1, t_0 + S)}{\partial \boldsymbol{W}_{i,j}(\tau)} \tag{61}$$

All quantities involved in the second term in Eq. 61 are within the current segment $(t_0 + 1, t_0 + S)$. Thus, this term can be efficiently computed using BPTT (see Appendix A.3).

The first term in Eq. 61 is trickier since we need to compute the gradients of the losses in the segment $(t_0 + 1, t_0 + S)$ w.r.t. weight variables from the segment $(1, t_0)$. This term can be computed through RTRL-style computations:

$$\sum_{\tau=1}^{t_0} \frac{\partial \mathcal{L}(t_0 + 1, t_0 + S)}{\partial \boldsymbol{W}_{i,j}(\tau)} = \sum_{\tau=1}^{t_0} \sum_{k=1}^{N} \frac{\partial \mathcal{L}(t_0 + 1, t_0 + S)}{\partial \boldsymbol{s}_k(t_0)} \times \frac{\partial \boldsymbol{s}_k(t_0)}{\partial \boldsymbol{W}_{i,j}(\tau)} \tag{62}$$

We can recognise that the first factor (we denote $\hat{\boldsymbol{\delta}}_k(t_0)$) can be computed as a function of $\boldsymbol{\delta}(t_0 + 1)$ (see the definition of $\boldsymbol{\delta}$ in Eq. 54) already computed by BPTT to compute the second term of Eq. 61 above (almost as $\boldsymbol{\delta}_k(t_0)$ through the backward recursion of Eq. 56 but without the first term),

$$\hat{\boldsymbol{\delta}}_k(t_0) = \frac{\partial \mathcal{L}(t_0 + 1, t_0 + S)}{\partial \boldsymbol{s}_k(t_0)} = \sum_{t'=t_0+1}^{t_0+S} \frac{\partial \mathcal{L}(t')}{\partial \boldsymbol{s}_k(t_0)} = \sum_{n=1}^{N} \boldsymbol{\delta}_n(t_0 + 1) \boldsymbol{R}_{n,k} \sigma'(\boldsymbol{s}_k(t_0)) \tag{63}$$

Eq. 62 becomes:

$$\sum_{\tau=1}^{t_0} \frac{\partial \mathcal{L}(t_0 + 1, t_0 + S)}{\partial \boldsymbol{W}_{i,j}(\tau)} = \sum_{\tau=1}^{t_0} \sum_{k=1}^{N} \hat{\boldsymbol{\delta}}_k(t_0) \frac{\partial \boldsymbol{s}_k(t_0)}{\partial \boldsymbol{W}_{i,j}(\tau)} = \sum_{k=1}^{N} \hat{\boldsymbol{\delta}}_k(t_0) \sum_{\tau=1}^{t_0} \frac{\partial \boldsymbol{s}_k(t_0)}{\partial \boldsymbol{W}_{i,j}(\tau)} \tag{64}$$

where we assume the last term $\mathbf{W}_{k,i,j}(t_0) = \sum_{\tau=1}^{t_0} \dfrac{\partial \boldsymbol{s}_k(t_0)}{\partial \boldsymbol{W}_{i,j}(\tau)}$ is computed while processing the previous segment; this completes the computation of the gradient in Eq. 59, i.e., for the last segment, this terminates the algorithm. If the current segment is not the last one, we have to compute $\mathbf{W}_{k,i,j}(t_0 + S)$ needed in the next iteration. This can be achieved by the following forward recursion:

$$\mathbf{W}_{k,i,j}(t_0 + S) = \sum_{\tau=1}^{t_0+S} \frac{\partial \boldsymbol{s}_k(t_0 + S)}{\partial \boldsymbol{W}_{i,j}(\tau)} = \sum_{\tau=1}^{t_0} \frac{\partial \boldsymbol{s}_k(t_0 + S)}{\partial \boldsymbol{W}_{i,j}(\tau)} + \sum_{\tau=t_0+1}^{t_0+S} \frac{\partial \boldsymbol{s}_k(t_0 + S)}{\partial \boldsymbol{W}_{i,j}(\tau)} \tag{65}$$

$$= \sum_{\tau=1}^{t_0} \sum_{l=1}^{N} \frac{\partial \boldsymbol{s}_k(t_0 + S)}{\partial \boldsymbol{s}_l(t_0)} \frac{\partial \boldsymbol{s}_l(t_0)}{\partial \boldsymbol{W}_{i,j}(\tau)} + \sum_{\tau=t_0+1}^{t_0+S} \sum_{l=1}^{N} \frac{\partial \boldsymbol{s}_k(t_0 + S)}{\partial \boldsymbol{s}_l(\tau)} \frac{\partial \boldsymbol{s}_l(\tau)}{\partial \boldsymbol{W}_{i,j}(\tau)} \tag{66}$$

We introduce the notation $\boldsymbol{H}_{k,l}(\tau) = \dfrac{\partial \boldsymbol{s}_k(t_0 + S)}{\partial \boldsymbol{s}_l(\tau)}$ for $\tau \in \{t_0, t_0 + 1, ..., t_0 + S\}$ which can be computed during BPTT in the segment $(t_0 + 1, t_0 + S)$ through the following backward recursion:

$$\boldsymbol{H}_{k,l}(\tau) = \sigma'(\boldsymbol{s}_l(\tau)) \sum_{m=1}^{N} \boldsymbol{H}_{k,m}(\tau + 1) \boldsymbol{R}_{m,l} \tag{67}$$

with $\boldsymbol{H}_{k,l}(t_0 + S) = 1$ if $k = l$ and 0 otherwise. Now Eq. 66 can be expressed as:

$$\mathbf{W}_{k,i,j}(t_0 + S) = \sum_{\tau=1}^{t_0} \sum_{l=1}^{N} \boldsymbol{H}_{k,l}(t_0) \frac{\partial \boldsymbol{s}_l(t_0)}{\partial \boldsymbol{W}_{i,j}(\tau)} + \sum_{\tau=t_0+1}^{t_0+S} \sum_{l=1}^{N} \boldsymbol{H}_{k,l}(\tau) \frac{\partial \boldsymbol{s}_l(\tau)}{\partial \boldsymbol{W}_{i,j}(\tau)} \qquad (68)$$

$$= \sum_{l=1}^{N} \boldsymbol{H}_{k,l}(t_0) \mathbf{W}_{l,i,j}(t_0) + \sum_{\tau=t_0+1}^{t_0+S} \boldsymbol{H}_{k,i}(\tau) \boldsymbol{x}_j(\tau) \qquad (69)$$

We remind that $\dfrac{\partial \boldsymbol{s}_l(\tau)}{\partial \boldsymbol{W}_{i,j}(\tau)}$ (in the last term of Eq. 68) can be simply computed by differentiating

Eq. 2 ($\dfrac{\partial \boldsymbol{s}_l(\tau)}{\partial \boldsymbol{W}_{i,j}(\tau)} = \boldsymbol{x}_j(\tau) \mathbb{1}_{l=i}$). We cache the following quantities to process the next segment:

$\boldsymbol{s}_i(t_0 + S), \mathbf{W}_{k,i,j}(t_0 + S)$, and $\dfrac{\partial \mathcal{L}(1, t_0 + S)}{\partial \boldsymbol{W}_{i,j}}$ for all $i, k \in \{1, ..., N\}$ and $j \in \{1, ..., D\}$.

The space complexity of this algorithm is $O(N^3)$, i.e., the same as in RTRL, because of $\mathbf{W}_{k,i,j}(t_0)$ to be stored. The time complexity is less than RTRL since it requires the RTRL-like computation of Eq. 69 only once per segment, i.e., only every $S$ steps. Assuming $S \sim O(N)$, the per-step time complexity is reduced to $O(N^3)$. In addition, this algorithm allows us to leverage parallel computation over the time dimension (through BPTT; even though the BPTT part is anyway less expensive than the RTRL part). This hybrid algorithm is conceptually more attractive than the pure RTRL for scenarios where RTRL's fully online learning property is not crucial, e.g., training of language models (as discussed in Sec. 5 and Appendix B.1) or actor-critic methods using a large $M$ (Sec. 3.2).

### A.5   feLSTM ARCHITECTURE

Here we introduce 'feLSTM' architecture that is an LSTM variant involving full recurrence, which is the closest to our eLSTM. We introduce this architecture so that we can compare our "eLSTM + exact RTRL" system with some fully recurrent RNN trained with an RTRL approximation in Table 2.

In fact, careful readers might note that, strictly speaking, such a comparison is not "fair" in the sense that we change both the base RNN architecture and the learning algorithm, given that the choice of the base RNN architecture itself influences the performance: we have already shown that eLSTM outperforms LSTM on these tasks (Table 2). To minimize this architectural discrepancy, our fully recurrent architecture, "feLSTM," is obtained by minimally modifying the eLSTM architecture.

"feLSTM" is obtained by replacing the element-wise recurrence of eLSTM in Eq. 5 by:

$$\boldsymbol{f}(t) = \sigma(\boldsymbol{F}\boldsymbol{x}(t) + \boldsymbol{W}^f \boldsymbol{c}(t-1)) \quad ; \quad \boldsymbol{z}(t) = \tanh(\boldsymbol{Z}\boldsymbol{x}(t) + \boldsymbol{W}^z \boldsymbol{c}(t-1)) \qquad (70)$$

where $\boldsymbol{W}^f \in \mathbb{R}^{N \times N}$ and $\boldsymbol{W}^z \in \mathbb{R}^{N \times N}$ are trainable weight *matrices*. All other equations remain the same as in eLSTM. Because of this full recurrence, RTRL for this model is not tractable. This allows us to compare "eLSTM + exact RTRL" v. "feLSTM + approximate RTRL (SnAp-1; Menick et al. (2021))" in Table 2 while keeping the base RNN architecture as close as possible.

## B   EXPERIMENTAL DETAILS, EXTRA RESULTS AND REMARKS

### B.1   DIAGNOSTIC TASKS

**Copy task.**   Since the main focus of this work is to evaluate RTRL-based algorithms beyond diagnostic tasks, we only conduct brief experiments on a classic diagnostic task used in recent RTRL research work focused on approximation methods (Tallec & Ollivier, 2018; Mujika et al., 2018; Benzing et al., 2019; Menick et al., 2021; Silver et al., 2022): the copy task. Let $\ell$ be a positive integer. Sequences in the copy task have a length $2\ell$ where the $\ell$ first symbols are either 0 or 1, and all others are #. The task is to sequentially read symbols in such a sequence, and to output the binary pattern presented in the first part of the sequence when reading the sequence of # in the second part. We conduct experiments with $\ell = \{50, 500\}$. Unlike prior work (see, e.g., Benzing et al. (2019)), we do not introduce any curriculum learning; we train models from scratch on the entire dataset containing sequences of length ranging from 2 to $2\ell$. Note that we do not conduct much of a hyper-parameter

search; we find that for this task, trying a few values is enough to find configurations that achieve 100% accuracy. The corresponding hyper-parameters are as follows. Respectively for $\ell = (50, 500)$, we use a learning rate of $(1e^{-4}, 3e^{-5})$, a batch size of $(512, 128)$, and a hidden layer size of $(1024, 2048)$. We apply a gradient norm clipping of $1.0$, and use the Adam optimiser. The model has no input embedding layer; one-hot representations of the input symbols are directly fed to the recurrent eLSTM layer (Sec. 3.1). Since our RTRL algorithm (Sec. 3.1) requires no approximation, and the task is trivial, we achieve 100% accuracy provided that the RNN size is large enough and that training hyper-parameters are properly chosen. We confirm this for sequences with lengths of up to 1000.

## B.2 REMARKS AND LIMITATIONS

**Remarks on computational capabilities of eLSTM.**   As noted in the introduction, eLSTM has certain limitations in terms of computational capabilities. In fact, certain theoretical results/limitations known for one-layer Quasi-RNN (Merrill et al., 2020) directly apply to eLSTM (and other element-wise recurrent NNs). While we do not observe any empirical limitations on the main tasks of this work (Sec. 4), we find one algorithmic task on which eLSTM is not successful (at least we failed to find successful configurations), which is the code execution task (Fan et al., 2020; Irie et al., 2021). While we refer to the corresponding papers for the task description, this task requires the model to maintain dynamically changing values of a certain number of variables. We are not successful at finding any configuration of our eLSTM that achieves 100% sequence-level accuracy on this task, while this is straightforward with the standard fully recurrent LSTM (see also Irie et al. (2023)).

**Remarks on language modelling and supervised learning.**   Another popular "diagnostic" task used in recent RTRL research is small-scale language modelling (see, e.g., Benzing et al. (2019)). However, as we already discuss in Sec. 5, in practice, language modelling may not be the best target application of RTRL in modern deep learning (apart from test-time online adaptation, i.e., dynamic evaluation (Mikolov et al., 2010; Krause et al., 2018), maybe). While we also conduct brief language modelling experiments using the Penn Treebank dataset (as in Benzing et al. (2019)) in our preliminary studies, we do not find any interesting result (beyond tiny perplexity improvements) that is worth being reported here. Unfortunately, our 1-layer architecture is too limited to perform well on modern language modelling tasks (dominated by deep models/Transformers) or any other meaningful supervised tasks at scale. If we restrict ourselves to 1-layer models, the task has to be limited to a toy setting similar to those used in prior works, contradicting our original objective of "going beyond diagnostic tasks."

As a general note, for RTRL to scale beyond toy tasks in supervised learning, it would require an optimised implementation such as a custom CUDA kernel (to be competitive with today's highly optimised TBPTT implementations). This would require further engineering efforts (but also some thoughts on the algorithm too: in Appendix A.4 we mention the RTRL-BPTT hybrid as a more promising RTRL variant for supervised learning), which we leave for future work.

## B.3 DMLAB

**Experimental Details.**   Here we provide details of our DMLab experiments.   We use two memory tasks `rooms_select_nonmatching_object` and `rooms_watermaze`, and one reactive task `room_keys_doors_puzzle` in Sec. 4.   For the corresponding task descriptions and examples of game screenshots, we refer to `https://github.com/ deepmind/lab/blob/master/game_scripts/levels/contributed/dmlab30/ README.md#select-non-matching-object`. We use observations based on DMLab's `RGB_INTERLEAVED`. Our base model architecture is IMPALA's deep variant (Espeholt et al., 2018). Action space discretisation is also based on IMPALA (Espeholt et al., 2018). Note that, unlike our work, some prior work (e.g., R2D2 (Kapturowski et al., 2019)) use some "improved" versions of the action space discretisation (Hessel et al., 2019). Regarding the reward clipping, no sophisticated asymmetric clipping (Espeholt et al., 2018) is used but the simple $[-1, 1]$ clipping. Training hyper-parameters/configurations are listed in Table 4. For the memory tasks, pre-training (Sec. 4) is done using TBPTT with $M = 100$. Our implementation is based on `torchbeast` (Küttler et al., 2019). We run three main training runs. Each run requires about a day of training on a V100 GPU (this is also the case with ProcGen and Atari). Also note that DMLab is CPU intensive. For evaluation, we first evaluate the final model checkpoint on three sets of 100 test episodes each;

Table 3: Scores on two memory environments of **DMLab-30** for $M = 10$ and $M = 50$.

| $M$ | Architecture | Recurrence | Learning Algorithm | select_nonmatch | watermaze |
|---|---|---|---|---|---|
| $M = 50$ | LSTM | full | TBPTT | $61.6 \pm 0.6$ | $17.2 \pm 3.0$ |
| | feLSTM | full | TBPTT | $61.4 \pm 0.3$ | $35.2 \pm 3.8$ |
| | eLSTM | element-wise | TBPTT | $61.4 \pm 0.5$ | $44.5 \pm 1.5$ |
| | feLSTM | full | SnAP-1 | $53.5 \pm 0.6$ | $24.2 \pm 2.2$ |
| | eLSTM | element-wise | RTRL | $62.0 \pm 0.4$ | $52.3 \pm 1.9$ |
| $M = 10$ | LSTM | full | TBPTT | $9.3 \pm 0.9$ | $15.3 \pm 2.0$ |
| | feLSTM | full | TBPTT | $51.1 \pm 1.5$ | $11.4 \pm 1.5$ |
| | eLSTM | element-wise | TBPTT | $54.5 \pm 1.1$ | $15.8 \pm 0.9$ |
| | feLSTM | full | SnAP-1 | $49.2 \pm 0.3$ | $24.7 \pm 3.6$ |
| | eLSTM | element-wise | RTRL | $61.8 \pm 0.5$ | $40.2 \pm 5.6$ |

resulting in three mean test scores for each training run. We average these scores to obtain a single mean score for each training run. The final number we report is the mean and standard deviation of these scores across three training runs.

**Extra Results.** To show that RTRL does not have any unexpected negative impacts in a mostly reactive task, we also test our system on one environment of DMLab-30, room_keys_doors_puzzle, which is categorised as a reactive task according to Parisotto et al. (2020). We train with $M = 100$ for 100 M steps (with the action repeat of 4) from scratch. The mean episode length is about 450 steps. Table 5/rightmost column shows the scores. Effectively, all feedforward, TBPTT, and RTRL systems perform nearly the same (at least within 100 M steps/400 M frames). We note that these scores are comparable to the one reported by the original IMPALA (Espeholt et al., 2018) which is 28.0 after training on 1 B frames, which is much worse than the score reported by Kapturowski et al. (2019) for IMPALA trained using 10 B frames ($54.6$).

Table 3 is similar to Table 2 but for $M = 10$ and $M = 50$.

**Remarks on Online Learning.** A potential benefit of small $M$ is *more online* learning which could be more sample efficient. At least on DMLab rooms_select_nonmatching_object (Figure 1a vs 1d), we do observe some sample efficiency benefits of RTRL with a small $M$: at about 2.5 M steps, RTRL with $M = 10$ achieves a score over 50, while the one with $M = 100$ is below 40 at the same number of steps. However, if the goal is to obtain the best performing model, our conclusion is that a too small $M$ is indeed not a good idea (more online learning deteriorates the performance, likely due to sensitivity staling). We note that it is rare that we can conduct such experiments using *exact* RTRL without worrying about the approximation quality, which is a benefit of our framework.

**Practical Computational Costs.** While our code is not optimised to fully leverage the computational advantages of RTRL, here we provide some insightful numbers regarding the computational costs. Training an eLSTM-RTRL with hidden size $N$=512, batch size 32 and $M$=100, using the frozen vision module on rooms_watermaze requires about 5 GB GPU memory (corresponding to our experimental setting of Table 1). This is comparable to that of TBPTT for $M$=100; in fact the memory requirements of TBPTT for $M \in \{10, 50, 100, 300\}$ are respectively: $\{1.4, 2.9, 4.8, 12\}$ GB (the corresponding empirical law is 0.038 * $M$ + 1 GB). Therefore, in order to compute the untruncated gradients as in RTRL using TBPTT for rooms_watermaze (simply assuming that $M$=1000 is needed, as 1000 is the average episode length for rooms_watermaze), the corresponding memory requirement is greater than 39 GB. In contrast, RTRL using only 5 GB allows to compute untruncated gradients and yields good performance (Table 1).

Regarding the actual wall clock time (measured on V100), TBPTT/PyTorch-backward/automatic-differentiation is a bit faster (1900 steps per second) than our RTRL/custom-forward-differentiation (1250 steps per second) but again, our implementation is not professionally optimised, and leaves much room for improvements (formally, the time complexity of $O(MN^2)$ is the same for TBPTT and RTRL applied to our eLSTM, but RTRL requires some extra overheads such as copying large sensitivity matrices as extra recurrent states).

Table 4: Hyper-parameters for RL experiments. Parameters at the bottom are common to all settings (which are essentially taken from the Atari configuration of Espeholt et al. (2018)).

| Parameters | DMLab | Atari | Procgen |
|---|---|---|---|
| Input image dimension | 3x96x72 | 1x84x84 | 3x64x64 |
| 4-frame stack | No | Yes | No |
| Action repeat | 4 | 4 | No |
| RNN dimension | 512 | 256 | 256 |
| Number of IMPALA actors | | 48 | |
| Discount | | 0.99 | |
| Learning rate | | 0.0006 | |
| Batch size | | 32 | |
| Gradient clipping | | 40 | |
| Loss scaling factors (baseline, entropy) | | (0.5, 0.01) | |
| RMSProp (alpha, epsilon, momentum) | | (0.99, 0.01, 0) | |

## B.4 PROCGEN

All our experimental settings in ProcGen environments are the same as for DMLab described above, except that no action repeat is used (see Table 4 for a comprehensive overview). Following (Cobbe et al., 2020), we use 500 levels for training.

Most ProcGen environments are solvable using a feedforward policy even without frame-stacking (Cobbe et al., 2020). Note that there is a so-called *memory-mode* for certain games, making the task partially observable by making the world bigger, and restricting agents' observations to a limited area around them. However, as has been already reported in Paischer et al. (2022), we confirm that even in these POMDP settings, both the feedforward and LSTM baselines perform similarly. In our experiments, *Chaser* is the only ProcGen environment where we observe *clear* benefits of recurrent policies. In our preliminary studies, we also test *Caveflyer*/hard, *Caveflyer*/memory, *Dodgeball*/memory, *Jumper*/hard, *Jumper*/memory, and *Maze*/memory. However, in these environments, we do not observe any notable benefits of LSTM-based policies over the feedforward baseline, except some improvements in *Dodgeball*/memory mode: $10.1 \pm 0.4$ (feedforward) vs. $12.7 \pm 0.6$ (LSTM) on the training set.

## B.5 ATARI

All our experimental settings in the Atari environments are the same as for DMLab described above, except that 4-frame stacking is used (see also Table 4), and that we use 5 sets of 30 episodes each for evaluation. As stated in the main text, our selection of five environments follows Kapturowski et al. (2019). Regarding other tasks, we also consider *Solaris* from Atari, as Kapturowski et al. (2019) find longer BPTT spans useful for this task. However, they achieve such improvements by training for 10 B environment frames, which are beyond our compute budget (800 M frames is our reasonable number); we tried up to 4 B steps, but without observing benefits of RTRL. Similarly, no real benefit of RTRL is observed on *Skiing* within this number of steps (except that RTRL and Feedforward agents immediately achieve a score around $-8987$ but remain stuck there forever, while the score of TBPTT-based agents gradually increases but only until circa $-17418$). Other scores are reported in Table 5.

Table 5: Scores on Atari and DMLab-reactive (`rooms_keys_doors_puzzle`) environments.

| | Breakout | Gravitar | MsPacman | Q*bert | Seaquest | keys_doors |
|---|---|---|---|---|---|---|
| Feedforward | $234 \pm 12$ | $1084 \pm 54$ | $3020 \pm 305$ | $7746 \pm 1356$ | $4640 \pm 3998$ | $\mathbf{26.6} \pm 1.1$ |
| TBPTT | $\mathbf{305} \pm 29$ | $1269 \pm 11$ | $\mathbf{3953} \pm 497$ | $11298 \pm 615$ | $12401 \pm 1694$ | $26.1 \pm 0.4$ |
| RTRL | $275 \pm 53$ | $\mathbf{1670} \pm 358$ | $3346 \pm 442$ | $\mathbf{12484} \pm 1524$ | $\mathbf{12862} \pm 961$ | $26.1 \pm 0.9$ |

