# OpenReview forum: "Exploring the Promise and Limits of Real-Time Recurrent Learning"
_ICLR.cc/2024/Conference — ICLR 2024 poster_

### Official Review · Reviewer_cmdn · 2023-10-16

**Soundness:** 3 good
**Presentation:** 3 good
**Contribution:** 3 good
**Rating:** 6
**Confidence:** 3

**Summary:**

The authors explore the practicality of RTRL as a method for learning in RNNs. In particular, the authors consider an RNN architecture whereby RTRL is relatively cheap to employ, and demonstrate its advantages over standard truncated BPTT on a range of reinforcement learning tasks.

**Strengths:**

- paper is generally well written
- though RTRL has now been around for some decades, its practical applications remain unclear. Towards this end the authors do offer interesting results which should be interesting to the broader ML community
- the RNN architecture based on element-wise application the authors use, though not novel itself, offers an interesting avenue for making RTRL more practical

**Weaknesses:**

- One concern I have is that the authors do not convincingly demonstrate the benefit of their model compared to the status quo, which is a fully connected RNN + truncated BPTT. In terms of complexity, the authors do reduce the memory/computational cost on pointwise RNNs to order N^2 which is certainly better than RTRL on standard RNNs, but this is still as expensive as BPTT on a fully connected RNN. In fact, for a pointwise RNN I wonder if BPTT is in fact of order N (as the forward pass?) - so still cheaper than RTRL - but I may be incorrect there. In terms of performance, the authors show that eLSTM + RTRL outperforms truncated BPTT on all RNN arhitectures for one task (watermaze) which is impressive, but I would have liked to seen this for more tasks; e.g. can RTRL outperform fully connected RNN + BPTT on any of the 5 Atari tasks in Fig 2?
- Related to the above, it is not clear whether online learning has any intrinsic benefits for performance. The authors note that online learning "allows for updating weights immediately after consuming very new input", but in practice the weights are only updated at the same rate as truncated BPTT (at rate M) to avoid 'sensitivity matrix staling'. I appreciate this is a difficult question to answer, and I believe this shortcoming is mentioned as the last presented limitation, but I think this deserves more direct clarification/exploration given the subject of this work.
- All main results presented are with respect to performance in reinforcement learning (RL) tasks, but what about supervised learning tasks? e.g. sequence modeling in language. The authors do use the 'copy task', which is a supervised learning task, as a diagnostic task, but I would have liked to see comparisons with BPTT for hard supervised learning tasks. Or at least a rational for only considering RL tasks
- The connection to biology and the 'memory' trace (e.g. Eqs 12-14) is interesting but not presently quite vague/weak. Some references to biological evidence for such traces (e.g. eligibility traces) would be advised.
- Given this is a paper which explores the practical/emperical outcomes of RTRL, I would advise a more thorough analysis of the ideal conditions for RTRL. For example, why do the authors believe RTRL is only valuable in some of the 5 Atari environments in Fig 2? For what types of task would RTRL be ill-advised? The space for technical description and equations in the main text is significant (e.g. equations 8-14) and in general I believe this should be in part sacrificed (to the Appendix) for more intepretability of the results.

**Questions:**

- The authors suggest that their work shows tractable RTRL in Quasi-RNN (Bradbury et al. 2017) and Simple Recurrent Units (Lei et al. 2018). I understand this for the latter, since as far as I understand the eLSTM is just a one-layer instance, but I don't understand this for Quasi-RNN. Doesn't this architecture use convolutions? Does that also mean that RTRL has a space/time complexity of N^2?
- The authors state that, "in practice, it is known that M > 1 is crucial (for TD learning of the critic) for optimal performance". Is this true? My belief was that one-step TD learning was frequently employed. Is there a relevant citation?
- How were the hyperparameters (Table 5) selected?
Typos:
- Mozer (1989; 1991) already explore an RNN -> Mozer (1989; 1991) already explored/s an RNN
- several modern RNN architectures such Quasi-RNN -> several modern RNN architectures such as Quasi-RNN

---

> ### Author Response · Authors · 2023-11-15
> **Response to Reviewer cmdn, part 1/2**
>
> We thank the reviewer for valuable time reviewing this work.
> We noted that several points listed under “weakness” are actually clarification questions.
> Answers to many of these questions can be actually found in the paper, either in the main text or in the appendix. We are happy to help the reviewer find the answers as follows. We also appreciated that the reviewer reflected her/his doubts through a confidence score of 3.
>
> > *One concern I have is that the authors do not convincingly demonstrate the benefit of their model compared to the status quo, which is a fully connected RNN + truncated BPTT. In terms of complexity, the authors do reduce the memory/computational cost on pointwise RNNs to order N^2 which is certainly better than RTRL on standard RNNs, but this is still as expensive as BPTT on a fully connected RNN. In fact, for a pointwise RNN I wonder if BPTT is in fact of order N (as the forward pass?) - so still cheaper than RTRL - but I may be incorrect there.*
>
> No, that is not correct: BPTT for eLSTM remains the same as the standard RNN. We had also provided a review/background material on BPTT in Appendix A.3 in case this might be helpful.
>
> > *In terms of performance, the authors show that eLSTM + RTRL outperforms truncated BPTT on all RNN arhitectures for one task (watermaze) which is impressive, but I would have liked to seen this for more tasks; e.g. can RTRL outperform fully connected RNN + BPTT on any of the 5 Atari tasks in Fig 2?*
>
> Yes, eLSTM + RTRL outperforms the LSTM + TBPTT baseline on 3 / 5 tasks (Gravitar, Q*Bert, Seaquest) and perform similarly on the two others. The scores (corresponding to Fig 2) were provided in Table 5 in the appendix. That said, the most important result is that of Watermaze which is the hardest task we have in terms of memory.
>
> > *Related to the above, it is not clear whether online learning has any intrinsic benefits for performance. The authors note that online learning "allows for updating weights immediately after consuming very new input", but in practice the weights are only updated at the same rate as truncated BPTT (at rate M) to avoid 'sensitivity matrix staling'.*
>
> First of all, we’d like to clarify that as we wrote in Sec 3.2 (last sentence): *“The main focus of this work is to evaluate learning with untruncated gradients, rather than the potential for online learning.“* That said, a potential benefit of *more online* learning is to be more sample efficient. At least on DMLab Select-Non-Matching-Object (Figure 1 (a) vs (d)), we do observe some sample efficiency benefits of RTRL with a small M: at x = 2.5M steps, RTRL with M=10 achieves a score over 50, while the one with M=100 is below 40 at the same number of steps.
> However, if the goal is to obtain the best performing model, our conclusion is that a too small M is indeed not a good idea (more online learning deteriorates the performance, likely due to sensitivity staling).
>
>
> > *All main results presented are with respect to performance in reinforcement learning (RL) tasks, but what about supervised learning tasks? e.g. sequence modeling in language. The authors do use the 'copy task', which is a supervised learning task, as a diagnostic task, but I would have liked to see comparisons with BPTT for hard supervised learning tasks. Or at least a rational for only considering RL tasks*
>
> We agree we all want to see RTRL applied to hard supervised learning tasks or any tasks in general. But that is the very challenge of the RTRL research. This is exactly what we discuss in the 3rd and 4th paragraph of Appendix B1 starting from *“Remarks on language modelling and supervised learning”*. There, we essentially explain that (1) with the one-layer constraint, it is hard to obtain competitive models on meaningful supervised learning tasks (and alike), e.g., language modelling, and (2) TBPTT implementations used for supervised sequence learning tasks are highly optimised; we argue that, to be competitive with them, it requires both engineering (custom CUDA implementation) and algorithmic (referring to hybrid RTRL-BPTT algorithms we discuss in Appendix A.4) efforts.
> With RL tasks, (1) is not really an issue as we can still obtain good/competitive systems even with one-layer models (compared to the well-known IMPALA and R2D2 baselines).
> The second point (2) is not just a limitation of our work but an open challenge for RTRL itself, which we honestly discussed in the paper (reflecting the title of our paper). This is also our unique contribution, which can not be found in any prior work on RTRL. We really hope this perspective improves the reviewer’s view on the contribution of our work, currently rated as low as 2.
>
> > *The connection to biology and the 'memory' trace (e.g. Eqs 12-14) is interesting but not presently quite vague/weak.*
>
> Yes, we’ll add some references there for the final version. If the reviewer has any specific suggestions, that’ll be also welcome.

---

> > ### Author Response · Authors · 2023-11-15
> > **Response to Reviewer cmdn, part 2/2**
> >
> > > *Given this is a paper which explores the practical/emperical outcomes of RTRL, I would advise a more thorough analysis of the ideal conditions for RTRL. For example, why do the authors believe RTRL is only valuable in some of the 5 Atari environments in Fig 2? For what types of task would RTRL be ill-advised?*
> >
> > The answer is actually very simple: in any tasks that benefit from a recurrent policy, there is no reason not to use RTRL (if it is tractable as in our case). The benefit is expected to be larger if a task requires longer time-span credit assignments. Strictly speaking, there is no case where RTRL is *“ill-advised,”* but obviously, if a task is purely reactive, RTRL, and also any recurrent policy anyway, is unnecessary. Empirically, in all our experiments, we never observed any critically bad performance using RTRL.
> >
> > > *The authors suggest that their work shows tractable RTRL in Quasi-RNN (Bradbury et al. 2017) and Simple Recurrent Units (Lei et al. 2018). I understand this for the latter, since as far as I understand the eLSTM is just a one-layer instance, but I don't understand this for Quasi-RNN. Doesn't this architecture use convolutions? Does that also mean that RTRL has a space/time complexity of N^2?*
> >
> > No, that is not due to convolution. Quasi-RNN is not a purely convolutional network, it does have recurrence, and as the corresponding recurrence is element-wise, its complexity for RTRL reduces (for the one-layer case) following a similar derivation we provide for eLSTM. Also, our eLSTM is not just a *“one-layer instance”* of Simple Recurrent Units; they have similarities but not identical.
> >
> > > *The authors state that, "in practice, it is known that M > 1 is crucial (for TD learning of the critic) for optimal performance". Is this true? My belief was that one-step TD learning was frequently employed. Is there a relevant citation?*
> >
> > For example, you could check the A3C paper/Sec 3. (Mnih et al. 2016 “Asynchronous Methods for Deep Reinforcement Learning” https://arxiv.org/abs/1602.01783), and maybe also simply Sutton and Barto’s textbook/Chapter 7 in general, or any other recent papers using actor-critic methods (e.g., the baseline IMPALA paper we cite).
> >
> > > *Typos ...*
> >
> > Thank you very much! We’ll fix them.
> >
> > We believe we thoroughly answered all the questions asked by the reviewer and resolved all the main concerns.
> > If you find our response useful, please consider increasing the score. Thank you very much.

---

> > > ### Comment · Reviewer_cmdn · 2023-11-18
> > >
> > > Thank you to the authors for their detailed response and clarifications. It is clear to me the authors have worked hard in this area and demonstrate good expertise.
> > >
> > > Reanalysing Table 5, my one concern remains that there does not seem to be significant difference (looking at error bars) between RTRL with the proposed eLSTM architecture and standard LSTM + BPTT. I think the authors certainly make a convincing case that RTRL can be useful/competitive for long temporal credit assignment, but the argument that, as the authors respond, "in any tasks that benefit from a recurrent policy, there is no reason not to use RTRL" certainly seems an overly strong and premature statement to me.
> > >
> > > That said, I think it's a solid paper and I would be happy for it to be accepted. I will increase both my contribution score and overall score. May the area chair also note that my confidence score remains modest.

---

> ### Author Response · Authors · 2023-11-18
> **Thank you for the score update**
>
> Thank you very much for your response and the score update!
>
> Just a clarification regarding:
>
> > "in any tasks that benefit from a recurrent policy, there is no reason not to use RTRL" certainly seems an overly strong and premature statement to me.
>
> We wrote this because we literally cannot find any reason not to use RTRL *when* RTRL is tractable (which is the core problem in the general case; but not in our restricted 1-layer element-wise recurrence case). We do not have any reason not to use true/untruncated gradients when we can compute them. They either provide some large performance improvements (e.g., as in Watermaze) when they are useful, or close to no degradation (e.g., as in some of the Atari games) when they are irrelevant. We can not see any downside (apart from engineering efforts to efficiently implement a non-standard learning algorithm).

---

### Official Review · Reviewer_nFUx · 2023-10-18

**Soundness:** 4 excellent
**Presentation:** 3 good
**Contribution:** 2 fair
**Rating:** 6
**Confidence:** 4

**Summary:**

Real-time recurrent learning enables computing gradients in RNNs as the input sequence is processed. However, its complexity makes it impractical in realistic scenarios. In this work, the authors introduce a new version of the LSTM architecture for which RTRL is exact and cheap. They test the resulting algorithm in realistic RL tasks and show that it performs competitively to alternatives (TBPTT as well as approximations of RTRL on more standard LSTM architectures).

**After rebuttal**: After clarifications from the author, I would like to increase my score from 5 to 7. Given that 7 does not exist, I updated it to 6.

**Strengths:**

The paper studies in depth the known, yet underappreciated, fact that RTRL becomes much cheaper when recurrence is element-wise. The paper nicely cites early references containing similar insights. The empirical results show that it is better to change the architecture so that RTRL becomes exact, rather than approximating it to make it tractable.

**Weaknesses:**

I see three main weaknesses:

- Most of the experimental results (all except baseline comparison) seem to some extent obvious to me: the only difference between TBPTT to RTRL is an extended context length. It is thus not surprising that RTRL > TBPTT in all experiments. The context length for TBPTT is relatively short (it seems far smaller than the one modern GPUs would offer) so those experiments do not seem to answer the question:
    > In what scenarios would RTRL be able to replace BPTT in today’s deep learning?

    asked by the authors. Adding a baseline that uses as much context length as possible would help better contextualize the findings of the paper. To me, this is the main weakness of the paper, and fixing it may require substantial changes.

- The paper is missing a few relevant references:
    - Bellec et al. 2017: RTRL from a computational neuroscience perspective. One of the main contributions of the paper is to discuss how RTRL can be implemented biologically through eligibility traces.
    - Gupta et al. 2022: show that it is possible to parametrize some classes of linear RNNs (more precisely linear SSMs) diagonally, without losing too much performance. Orvieto et al. 2023, which is cited in this paper, build on this result.
    - Zucchet et al. 2023: show that spatial backpropagation combined with element-wise complex-valued recurrence enables learning multilayer networks with a performance close to the one of BPTT.
- The paper mentions the multilayer case as an important limitation but does not discuss any possible or existing solutions (layer-wise training of Javed et al. 2023 / spatial backpropagation of errors of Zucchet et al. 2023)

**Questions:**

The paper focuses on one layer, but mentions that multiple layers are key in deep learning. Would the experimental results presented here would benefit from RNNs with multiple layers? From the vision module being fine-tuned?

---

> ### Author Response · Authors · 2023-11-15
> **Response to Reviewer nFUx**
>
> We thank the reviewer for valuable time reviewing our work.
> While we also thank the reviewer for acknowledging certain strengths of our work, we also would like to draw the reviewer’s attention to other important aspects of our work which seem to be currently overlooked.
>
> > *Most of the experimental results (all except baseline comparison) seem to some extent obvious to me: the only difference between TBPTT to RTRL is an extended context length. It is thus not surprising that RTRL > TBPTT in all experiments.*
>
> We’d like to clarify that the goal of research on RTRL is not to *“surprise”* the readers; we all know that RTRL should be “better” but the whole challenge of any practical RTRL research is to build an actual system trained by RTRL that exhibits practical improvements.
>
> >  *The context length for TBPTT is relatively short (it seems far smaller than the one modern GPUs would offer) so those experiments do not seem to answer the question: In what scenarios would RTRL be able to replace BPTT in today’s deep learning? asked by the authors. Adding a baseline that uses as much context length as possible would help better contextualize the findings of the paper. To me, this is the main weakness of the paper, and fixing it may require substantial changes.*
>
> We’d like to clarify two important points here.
>
> First of all, our experiments on DMLab memory tasks (Table 1) already show two contrastive cases. On Select-Non-Matching-Object (where the mean episode length is short; about 100 steps) TBPTT with a large M is nearly equal to full BPTT: consequently we only observe very limited improvements with RTRL. On the other hand on Watermaze (where the mean episode length is about 1000 steps; in Appendix B.2 DMLAB, paragraph *“Practical Computational Costs”*, we discuss the corresponding memory requirements), clear benefits of RTRL are observed (Table 1). Overall, RTRL is effectively promising for tasks with long-span dependencies, and rather irrelevant otherwise.
>
> Now regarding the question “In what scenarios would RTRL be able to replace BPTT in today’s deep learning?”, the reviewer is missing one very important perspective of our work.
> We formulated this as a fundamental question that anybody interested in RTRL should ask.
> We do not intend to answer this question only through these experiments (not because something is missing in our experiments; but because the scope of this question is much broader). Rather, our goal is to encourage more broad thinking about RTRL in the scope of today’s (and maybe future) machine learning, going beyond the current focus on approximation methods.
> It is absolutely true that for many tasks (like Select-Non-Matching-Object here) full backpropagation spans are short enough to be fitted to current GPUs. This will be the case for even more tasks in the future with advances in the hardware. This is exactly what we discuss in Sec 5 paragraph *“Principled vs. practical solution”*. In a similar spirit, we also discuss the limitations and challenges of RTRL in supervised learning (3rd and 4th paragraphs in Appendix B1 starting from *“Remarks on language modelling and supervised learning”*).
>
> Overall, all these honest questions and discussions bring new perspectives on RTRL research which are completely ignored in prior work on RTRL. “Promise” and “limits” in our title reflect this.
> We argue that this is an important and novel contribution, which, we believe, merits more than a contribution score of 2.
>
> > *The paper is missing a few relevant references ...*
>
> Thank you very much for pointing out these papers. We'll cite them in our revision (except Javed et al. 2023 which is already cited), and add comments on existing efforts to deal with the multi-layer case.
>
> > *The paper focuses on one layer, but mentions that multiple layers are key in deep learning. Would the experimental results presented here would benefit from RNNs with multiple layers?*
>
> Yes, we observed slight improvements with deeper models (see the next paragraph) but here, we intentionally focused on RL tasks where we can still achieve good performance (compared to well-known IMPALA and R2D2 baselines) even with one layer. This does not contradict our statement on the general importance of deep models though. Also generally, a learning algorithm that is only exact and efficient for one-layer models is too limited.
>
> In our preliminary studies, we evaluated a baseline LSTM (trained with TBPTT) on DMLab-30 Watermaze with 1 or 2 layers: we found the 2-layer model (score 39.7 +- 4.6) to slightly outperform the 1-layer model (37.6 +- 5.3) while both of them underperform our 1-layer eLSTM (45.6 +- 4.7).
>
> We hope our response brings clarification on all the concerns, in particular the first one marked as *“the main weakness”*. The scope of our paper is different from previous RTRL papers, which is our unique contribution.
> If the reviewer finds our response convincing, we’d greatly appreciate it if it is reflected as a score change.

---

> ### Comment · Reviewer_nFUx · 2023-11-17
>
> I thank the authors for the clarifications.
>
> There was an important point that I didn't fully appreciate: when one wants to update parameters every $M$-steps one can only use context lengths of size at most $M$ with approaches rooted in BPTT when RTRL approaches can use, in principle, infinite context length. This is particularly important when sample efficiency is critical, as it is nicely demonstrated in the RL experiments. The authors may want to emphasize more that this will hold whatever the hardware memory size is. A posteriori, I think the discussion on the memory side distracted me from appreciating this argument that the authors make. Maybe quickly recalling the discussion in Section 3.2 in the "Principled vs. practical solution" would help other readers (the discussion in appendix B.1 on "Remarks on language modeling and supervised learning" would also help).
>
> Reviewer Pxbt asked about the expressivity of element-wise recurrence, here are a few references on the expressivity of linear element-wise recurrence that might interest the authors:
> - "Fading memory and the problem of approximating nonlinear operators with Volterra series", Boyd & Chua 1985
> - "Universal discrete-time reservoir computers with stochastic inputs and linear readouts using non-homogeneous state-affine systems", Grigoryeva & Ortega 2018
> - "On the Universality of Linear Recurrences Followed by Nonlinear Projections", Orvieto et al 2023
>
> Overall, I think the paper should be accepted and want to increase my score to 7. (EDIT: ICLR does not allow giving 7, so the updated score does not fully reflect my position)

---

> > ### Author Response · Authors · 2023-11-17
> > **Thank you for the score update**
> >
> > Thank you very much for your reply. We are glad to hear that our clarifications were successful.
> >
> > To be fully honest we'd have appreciated it if the reviewer could follow the ICLR scaling instead of using their own definition (likely the one from NeurIPS; essentially 7 there is 8 here). We hope that the AC will not miss this "conversion" step in the busy process of meta-reviewing. In any case, thank you very much for the score update!
> >
> > PS: Thank you for further expressivity references. The most relevant citation here seems to be Merrill et al. ACL 2020 which we already cite though. They deal with the practical computational capabilities that we really care about (in the spirit of Weiss et al. 2018 "On the practical computational power of finite precision RNNs for language recognition"; where we can e.g., distinguish between the capabilities of LSTM and GRU).

---

### Official Review · Reviewer_8WmR · 2023-10-31

**Soundness:** 4 excellent
**Presentation:** 4 excellent
**Contribution:** 3 good
**Rating:** 6
**Confidence:** 4

**Summary:**

This work studies RTRL method on element-wise recurrent neural networks, examined with the actor-critic policy gradient algorithm (R2AC) on reinforcement learning tasks. Although the eLSTM architecture was not first proposed by authors, the connection between element-wise recurrence and RTRL training was first pointed out. The eLSTM, RTRL and R2AC are examined together on several subsets of DMLab-30, ProcGen and Atari-2600 envrionments and outperforms the IMPALA and R2D2 baselines trained on 10B frames.

**Strengths:**

1. The study of RTRL on element-wise recurrence provides a clean examination condition to study the potential of RTRL learning algorithm. From the provided experimental results the RTRL outperforms TBPTT on eLSTM and other baselines on most of the tasks.
2. Very clear writing, easy to follow. It is a big strength that the authors provided a thorough analysis of the limitations including multi-layer RTRL, the practicality of RTRL etc.

**Weaknesses:**

1. Only one element-wise RNN instantiation, namely the eLSTM was provided for comparison. It would be better to show the generality of the method on some other element-wise RNN instantiations.
2. The experiments are conducted on reinforcement learning tasks, which is fine as it is stated clearly as the study goal. However, it would be much more convincing if it is shown with certain supervised learning tasks.

**Questions:**

1. The variances shown in Figure 2 across different Atari environments are not consistent. For example, in Gravitar the variance for RTRL is larger than the other two and it’s the opposite in Seaquest. But overall it seems RTRL has a larger variance. Could the authors provide some explanation or intuition for that?

---

> ### Author Response · Authors · 2023-11-15
> **Response to Reviewer 8WmR**
>
> We thank the reviewer for valuable time reviewing our work, and for many positive comments.
>
> > *Although the eLSTM architecture was not first proposed by authors, the connection between element-wise recurrence and RTRL training was first pointed out.*
>
> We’d like to clarify that we are not the first to point out that the complexity of RTRL reduces for certain element-wise recurrent NNs (we clarify this upfront in the introduction/3rd paragraph by citing Mozer’s work from the late 1980s). This is a “technically not new” but “largely ignored/unexplored” fact which we revisit in the modern context.
>
> > *Only one element-wise RNN instantiation, namely the eLSTM was provided for comparison. It would be better to show the generality of the method on some other element-wise RNN instantiations.*
>
> We had to make a decision given our limited compute resource. As we expressed in the main text Sec 3.1 (*“While one could further discuss myriads of architectural details (Greff et al., 2016), most of them are irrelevant to our discussion on the complexity reduction in RTRL; the only essential property here is that “recurrence” is element-wise.”*), all these existing element-wise RNN architectures are similar up to certain details. Therefore, instead of evaluating various architectures with small differences, we opted for conducting experiments across many tasks. We hope this clarifies the logic of our decision.
>
> > *The experiments are conducted on reinforcement learning tasks, which is fine as it is stated clearly as the study goal. However, it would be much more convincing if it is shown with certain supervised learning tasks.*
>
> This is a natural remark which we actually discuss in the paper (3rd and 4th paragraphs in Appendix B1 starting from *“Remarks on language modelling and supervised learning”*). There, we essentially comment that (1) with the one-layer constraint, it is hard to obtain competitive models on meaningful supervised learning tasks (and alike), e.g. language modelling, and (2) TBPTT implementations typically used for supervised sequence learning tasks are highly optimised; we argue that, to be competitive with them, it requires both engineering (custom CUDA implementation) and algorithmic (referring to hybrid RTRL-BPTT algorithms we discuss in Appendix A.4) efforts. This is not just a limitation of our work but an open challenge for RTRL itself, which we honestly discuss in the paper, reflecting the title of our paper. Such discussions are also our unique contributions, which can not be found in any prior work on RTRL. We really hope this perspective improves the reviewer’s view on the contribution of our work, currently rated as low as 2.
>
> > *The variances shown in Figure 2 across different Atari environments are not consistent. For example, in Gravitar the variance for RTRL is larger than the other two and it’s the opposite in Seaquest. But overall it seems RTRL has a larger variance. Could the authors provide some explanation or intuition for that?*
>
> Right, these variances do not have to be consistent. Our intuition is as follows. There are two key phenomena: discovering a key credit assignment within a trajectory, and encountering “good” trajectories containing such credit assignment.  RTRL should facilitate the former (thanks to untruncated gradients) as long as the latter is successful. The latter, experiencing good “trajectories” containing the relevant credit assignment paths, depends on the nature of the environment in particular. We speculate that it happens to be easy to encounter good trajectories in Seaquest, allowing RTRL to consistently find good strategies (low variance), while this does not seem to be the case for Gravitar where RTRL can yield good performance only when “good” trajectories are found (high variance). This may be a possible explanation of what we observe there.
>
> We hope our response improves the reviewer’s overall view of this work and its contributions in particular.
> If you find our response convincing, please consider increasing the score. Thank you very much.

---

> > ### Comment · Reviewer_8WmR · 2023-11-19
> >
> > Thanks for the authors. I have a better understanding of the unique contributions of this work now. I am increasing the contribution score and the confidence score.

---

### Official Review · Reviewer_PxbF · 2023-11-01

**Soundness:** 3 good
**Presentation:** 3 good
**Contribution:** 4 excellent
**Rating:** 8
**Confidence:** 4

**Summary:**

In this paper, the authors explore the advantages of real-time recurrent learning (RTRL) compared to truncated BPTT in many reinforcement learning settings scoped to memory tasks. They use a variant of LSTM with element-wise recurrence in a singe layer to make RTRL tractable, and show that RTRL does have an advantage over truncated BPTT in these scenarios. They also provide a discussion of the difficulty of using RTRL in multi-layer recurrent networks.

**Strengths:**

While RTRL is known to be computationally intractable in the general case, there has not been much prior work in exploring if specific architectural choices can help make exact RTRL more feasible and useful. Especially given the potential advantages RTRL could have due to its computational requirements being independent of sequence length unlike many other sequence models. So this work is very timely and relevant to the community, and the use of element-wise recurrence to make exact RTRL feasible is novel to my knowledge.

The authors demonstrate many tasks where RTRL with this architecture does better than truncated BPTT, which is very interesting given the very limited recurrence used. The discussion of multi-layer RTRL is also significant and useful to the community. Overall, the quality and clarity of the paper are high.

**Weaknesses:**

- The elementwise recurrence that the authors propose for eLSTM is very similar to that used in IndyLSTM [1], but a comparative discussion and citation is missing.
- Since exact RTRL computes exactly the same gradient as full BPTT, a specific analysis of why TBPTT rather than BPTT is used in each of the demonstrated tasks would be important to provide context (for e.g. in regards to memory requirements).
- A discussion/analysis of the computational expressivity of having restricted recurrent and its implications is missing.
- On a related note, a discussion (or even speculation) of why eLSTM works as well as it does given its limited recurrence would be useful.
- Certain combinations of baselines are missing: eLSTM with SnAP-1, full BPTT.
- The clarity of the experiments section could be improved. See below in "Questions".

[1] (Gonnet & Deselaers 2019) https://arxiv.org/abs/1903.08023

**Questions:**

- Why does IMPALA with TBPTT+eLSTM do better than standard IMPALA (with LSTM+TBPTT presumably?)
- Is having a small value of M beneficial for RTRL in any other way? Not clear what the motivation for studying the effect of M is.

## Clarity:
- The specific tasks the authors test on is not described anywhere in the paper (the Appendix points to a webpage). Having this would help a lot with understanding the tasks used in the paper.
- The differences between the baseline methods of various variants of IMPALA and R2D2 are not sufficiently described.

---

> ### Author Response · Authors · 2023-11-15
> **Response to Reviewer PxbF**
>
> We thank the reviewer for valuable time reviewing our work and for many very positive comments on the significance of this work. We respond to the remaining questions and concerns as follows:
>
> > *The elementwise recurrence that the authors propose for eLSTM is very similar to that used in IndyLSTM [1], but a comparative discussion and citation is missing.*
>
> Yes, the reviewer is right. We listed/cited many of them in Sec 3.1., including “IndLSTM” which even has almost the same name, but this one was indeed missing. Thank you for pointing this out; we’ll cite it.
> We emphasise that they are all “similar” but not identical, and eLSTM has a minimalistic design.
>
> > *Since exact RTRL computes exactly the same gradient as full BPTT, a specific analysis of why TBPTT rather than BPTT is used in each of the demonstrated tasks would be important to provide context (for e.g. in regards to memory requirements).*
>
> We primarily use TBPTT as it is the standard algorithm used to train recurrent policies in general. As we also wrote in the main text (paragraph “Principled vs. practical solution” in Sec. 5), in certain tasks such as DMLab Select-Non-Matching-Object, TBPTT with a large M is nearly full BPTT. Regarding the memory benefits of RTRL, we discuss its details in Appendix “B.2 DMLAB” paragraph “Practical Computational Costs.“
>
> > *A discussion/analysis of the computational expressivity of having restricted recurrent and its implications is missing.*
>
> We actually discuss this in Appendix “B.1 DIAGNOSTIC TASKS” (Page 22), paragraph “Remarks on computational capabilities of eLSTM.” In particular, we refer to Merrill et al. ACL 2020’s theoretical work on quasi-RNNs’ expressivity (which is a representative element-wise RNN). That said, we admit that this paragraph should be hard to find currently. We will add a sentence in the main text pointing to this appendix section so that people can find this (and we likely should make this an independent subsection instead of a paragraph).
>
> > *On a related note, a discussion (or even speculation) of why eLSTM works as well as it does given its limited recurrence would be useful.*
> > *Why does IMPALA with TBPTT+eLSTM do better than standard IMPALA (with LSTM+TBPTT presumably?)*
>
> One speculation is that element-wise recurrence is easier to learn as it does not involve any full matrix-vector multiplication for temporal/recurrent dependencies, making it more sample efficient. But this remains a tricky question.
>
> > *Certain combinations of baselines are missing: eLSTM with SnAP-1, full BPTT.*
>
> There should be some confusion here: “SnAP-1” and “full BPTT” are both learning algorithms which can only be used exclusively. Also, the combination “eLSTM with SnAP-1” does not exist: SnAP-1 is an approximation method for the fully recurrent models, which does not apply to eLSTM whose recurrence is element-wise. As far as we can tell, our current Table 2 is complete: it already covers all interesting/relevant cases.
>
> > *The clarity of the experiments section could be improved. See below in "Questions".*
> > *The specific tasks the authors test on is not described anywhere in the paper (the Appendix points to a webpage).*
>
> We originally skipped these descriptions as they will be very long given the number of tasks we cover. But if the reviewer thinks they are really necessary, we’ll be happy to add the corresponding task descriptions to the appendix.
> Regarding the full details about the external baseline models, we refer to the corresponding papers.
>
> > *Is having a small value of M beneficial for RTRL in any other way? Not clear what the motivation for studying the effect of M is.*
>
> A potential benefit of small M is *more online* learning which could be more sample efficient. At least on DMLab Select-Non-Matching-Object (Figure 1 (a) vs (d)), we do observe some sample efficiency benefits of RTRL with a small M: at x = 2.5M steps, RTRL with M=10 achieves a score over 50, while the one with M=100 is below 40 at the same number of steps.
> However, if the goal is to obtain the best performing model, our conclusion is that a too small M is indeed not a good idea (more online learning deteriorates the performance, likely due to sensitivity staling).
> Overall, studying/evaluating the effect of M here is natural. We also note that it is rare that we can conduct such experiments using the *exact* RTRL without worrying about the approximation quality, which is a clear benefit of our framework.
>
> We hope our response above brings clarifications to all the remaining questions. Otherwise, we’ll be happy to clarify more.

---

> ### Comment · Reviewer_PxbF · 2023-11-20
>
> Thank you for your responses and explanations -- they helped me understand your points better, and may be helpful for readers of the papers as well. Pointers in the main text to interesting analyses in the supplement would definitely be very helpful.
>
>
> >> Certain combinations of baselines are missing: eLSTM with SnAP-1, full BPTT.
>
> > There should be some confusion here: “SnAP-1” and “full BPTT” are both learning algorithms which can only be used exclusively. Also, the combination “eLSTM with SnAP-1” does not exist: SnAP-1 is an approximation method for the fully recurrent models, which does not apply to eLSTM whose recurrence is element-wise. As far as we can tell, our current Table 2 is complete: it already covers all interesting/relevant cases.
>
> My understanding of SnAP-1 is that it discards non-diagonal terms in the influence matrix update. So I'm not sure why it would not be possible to use this to train eLSTM.
>
> As for full BPTT, is the argument that since it will calculate exactly the same gradients as RTRL with M set to sequence length, it doesn't make sense to list it explicitly? What is the sequence length for the experiments in Table 1?
>
> >> The clarity of the experiments section could be improved. See below in "Questions". The specific tasks the authors test on is not described anywhere in the paper (the Appendix points to a webpage).
>
> > We originally skipped these descriptions as they will be very long given the number of tasks we cover. But if the reviewer thinks they are really necessary, we’ll be happy to add the corresponding task descriptions to the appendix. Regarding the full details about the external baseline models, we refer to the corresponding papers.
>
> What I had in mind is just single sentence conceptual descriptions of what the tasks are, but I'll defer to the authors on whether to put in the main text.

---

> > ### Author Response · Authors · 2023-11-20
> >
> > Thank you for your reply.
> >
> > > My understanding of SnAP-1 is that it discards non-diagonal terms in the influence matrix update. So I'm not sure why it would not be possible to use this to train eLSTM.
> >
> > Yes, the reviewer's understanding is correct, but the non-diagonal terms in that "influence" matrix are already zero in element-wise recurrent models. This is why we introduced an additional architecture 'feLSTM' (Table 2, Appendix A.5) which is a minimum modification to eLSTM with fully recurrent components that still benefit from the SnAP-1 approximation.
> >
> > > As for full BPTT, is the argument that since it will calculate exactly the same gradients as RTRL with M set to sequence length, it doesn't make sense to list it explicitly?
> >
> > Exactly.
> >
> > > What is the sequence length for the experiments in Table 1?
> >
> > The average episode lengths for Select-Non-Matching-Object and Watermaze are about 100 and 1000 respectively. So for the former, TBPTT with M=100 is nearly full BPTT. Regarding Watermaze, we discuss its high memory requirement for full BPTT
> > in Appendix B.2 DMLAB, paragraph “Practical Computational Costs”.
> >
> > > What I had in mind is just single sentence conceptual descriptions of what the tasks are, but I'll defer to the authors on whether to put in the main text.
> >
> > Thank you for this clarification.

---

> > > ### Comment · Reviewer_PxbF · 2023-11-20
> > >
> > > >> My understanding of SnAP-1 is that it discards non-diagonal terms in the influence matrix update. So I'm not sure why it would not be possible to use this to train eLSTM.
> > >
> > > > Yes, the reviewer's understanding is correct, but the non-diagonal terms in that "influence" matrix are already zero in element-wise recurrent models. This is why we introduced an additional architecture 'feLSTM' (Table 2, Appendix A.5) which is a minimum modification to eLSTM with fully recurrent components that still benefit from the SnAP-1 approximation.
> > >
> > > >> As for full BPTT, is the argument that since it will calculate exactly the same gradients as RTRL with M set to sequence length, it doesn't make sense to list it explicitly?
> > >
> > > > Exactly.
> > >
> > > Makes sense. Thanks for the clarification.

---

### Author Response · Authors · 2023-11-15
**General Response**

We thank all the reviewers for their valuable comments.

For the moment, we will keep the originally submitted PDF as is; we’ll apply all promised changes for the final version if the paper is accepted.

Overall, we believe that our response should resolve all main concerns raised by the reviewers. Please find our individual responses below.

---

### Meta-Review · Area_Chair_wwGd · 2023-12-06

**Metareview:**

This paper explores the practical potential and limitations of real-time recurrent learning (RTRL). RTRL is an alternative to backprop for calculating gradients through time, but the full version requires memory expensive computations. The authors show that when applied to single-layer recurrent networks with element-wise recurrence (which renders the memory requirements more tractable), RTRL can perform well on reinforcement learning tasks using an actor-critic approach. They also discuss the remaining practical limitations, such as applicability of RTRL for multi-layer networks.

The reviews for this paper were generally positive, with some concerns about the limited nature of the evaluations and the fairness of comparisons to truncated backprop through time. However, the authors did a good job in rebuttal of clarifying the goals of the study and articulating why the results are informative and of interest. The final scores were 8,6,6,6, placing this paper clearly above the 'accept' threshold.

**Justification For Why Not Higher Score:**

This paper is an interesting demonstration of how RTRL can be made usable, but it only does so for a very limited setting. Thus, I don't think it will be of succifient interest to the ICLR community for a spotlight.

**Justification For Why Not Lower Score:**

The reviewers agreed that this is a solid paper, and the scores were clearly above threshold.

---

### Decision · Program_Chairs · 2024-01-16

Accept (poster)